# A Mathematical Model for Efficient and Fair Resource Assignment in Multipath Transport

**Andreas Könsgen** *[ID]**, Md. Shahabuddin, Amanpreet Singh and Anna Förster** [ID]

Sustainable Communication Networks, University of Bremen, 28359 Bremen, Germany;
shahab@comnets.uni-bremen.de (M.S.); aps@comnets.uni-bremen.de (A.S.);
afoerster@comnets.uni-bremen.de (A.F.)
**\*** Correspondence: ajk@comnets.uni-bremen.de

**Abstract:** Multipath transport protocols are aimed at increasing the throughput of data flows as well as maintaining fairness between users, which are both crucial factors to maximize user satisfaction. In this paper, a mixed (non)linear programming (MINLP) solution is developed which provides an optimum solution to allocate link capacities in a network to a number of given traffic demands considering both the maximization of link utilization as well as fairness between transport layer data flows or subflows. The solutions of the MINLP formulation are evaluated w. r. t. their throughput and fairness using well-known metrics from the literature. It is shown that network flow fairness based capacity allocation achieves better fairness results than the bottleneck-based methods in most cases while yielding the same capacity allocation performance.

**Keywords:** multipath transport; fairness; linear programming

## 1. Introduction

The aim of transport-layer protocols is the reliable end-to-end transport of data. For best performance and resulting user satisfaction, Internet providers expect that transport protocols utilize links in an optimum way and avoid congestion in the network. Furthermore, protocols have to consider that links are in most cases shared by multiple users. Therefore, the available capacity on a shared link should be assigned to the users in a fair way so that no user has to starve. Fairness means that a transport protocol should respond to congestion notifications such as packet loss or increase of delay by reducing the traffic load injected into the network.

The most commonly used transport protocol in today's Internet is the Transmission Control Protocol (TCP) [1]. It is not only used by "classical" applications which rely on reliable transmissions—such as web, e-mail or file transfer—but also by some soft-real-time applications where delay is less critical, such as video streaming. In the context of TCP coexisting with other transport protocols on shared links, the term "TCP friendliness" means that a flow should not use a larger portion of the link capacity than a legacy TCP flow [2].

An extension for legacy TCP developed in recent years is Multipath TCP (MPTCP) [3] which makes use of multiple interfaces in the sender or receiver in contrast to legacy TCP, which is only able to use a single interface for a given data flow. Each interface connects the sender or receiver node to a different network so that packets are sent to the opposite station along different paths. Over each of the paths, a subset of the overall amount of packets is transported; this subset is called MPTCP subflow. The protocol stack hides details about the transport from the application, it behaves in a transparent way and appears to the application as a legacy connection.

An alternative transport-layer protocol developed in recent years which from the beginning has been designed for multihoming operation is the Stream Control Transmission Protocol (SCTP)

[4]. However, although multihoming means that alternative interfaces on a node and the resulting additional data paths are identified, they are only used for redundancy, so that an extension named Concurrent Multipath Transfer SCTP (CMT-SCTP) has been investigated [5] and is currently proposed as an Internet Engineering Task Force (IETF) draft [6].

It was pointed out that any transport-layer protocol, irrespective of whether it supports multipath, should coexist in a fair way with legacy TCP, so fairness has to be ensured when sharing resources with normal TCP flows. A resource can either be a link which is part of an end-to-end connection, or it can be the entire network. Moreover, when discussing fairness, different participants can be specified. On the one hand, there are data flows which transport the entire data of a stream or file. On the other hand, such a flow can be divided into subflows where each of them transports a subset of the data. The TCP-friendliness policy may require that an entire data flow may behave equivalently to a TCP connection, or each of the subflows should do so. This option of different resources and participants means that there is not "the" fairness as such, but there are different ways regarding how to interpret fairness.

Another aspect of multipath transport is that it can not only consider the entire *network* as a resource which should be fairly shared as described in Section 3.2.3 of this article, but also focus on individual links as the shared resource as shown in Sections 3.2.1 and 3.2.2 , respectively.

This paper discusses the theoretical analysis of resource assignment and fairness in the domain of multipath *transport*. In the multipath transport case, at least one of the end nodes of a connection has multiple interfaces, usually connected to different access networks, so that a data flow can be split at the transport layer. The analytical description of multipath transport requires mathematical formulation which avoids flow splitting "on the way"; it has to be restricted to the end nodes by respective constraints. The topic should not be confused with multipath *routing* where data is sent from a source node to a destination node as a single flow from the end-to-end view, whereas the flow splitting occurs in routers along the way between sender and receiver. Multipath transport is run between end-user devices as previously said. In this way, both methods complement each other. Furthermore, it was previously mentioned that multipath transport fairness can be performed not only inside the resource of an entire network, but also in an individual bottleneck link, whereas multipath routing always has the network as the scope.

Existing research about analytical modeling of multipath transport focuses on analytical modeling of the behaviour and stability of practical MPTCP congestion control [7–11]. Any practical transport-layer algorithm is however limited by the fact that it does not have knowledge about the internal states of the network; it takes estimation by observing the performance of the links. Furthermore, practical algorithms respond to changing link conditions, which results in a time-dependent behaviour. In contrast to this, the aim of our paper is looking at a network in "god view" in order to find the optimum resource assignment assuming perfect knowledge of the network topology in a static equilibrium. The considerations in our paper are independent from a particular algorithm; however, they consider fairness which is also inherent to the different practical algorithms as discussed in Sections 3.2.1–3.2.3. Another group of publications discusses the aforementioned fairness in multipath routing where the survey [12] gives an overview and some further examples including [13–16]. Our paper differs, as already mentioned, by considering fairness between end-to-end connections instead of routes in a network. Further publications discuss legacy (single-path) TCP where various methods are summarized in three surveys [17–19]. In addition, for UDP transport, an application-layer congestion control scheme has been proposed to support fairness [20]. Finally, authors have investigated different practical implementations of MPTCP; these references are summarized in Sections 3.2.1–3.2.3 where they are given as examples of how different fairness methods were realized.

The contribution of this paper is (1) presenting formal definitions of a network and the different transport-layer fairness methods based on selecting resources and participants; (2) based on these definitions, designing a novel mathematical method based on mixed-integer (non)linear programming

(MINLP) for the optimum assignment of data flows to physical paths across the network; and (3) evaluating the developed mathematical methods using 30 example scenarios.

The proposed method should consider a trade-off between high link utilization and fair share as discussed in the previous paragraphs. Since fairness is an important aspect of the resource assignment problem, well-known fairness methods from the literature are used to measure the fairness in a quantitative way. The methods can serve as a benchmark to evaluate practical resource assignment methods for multipath transport protocols.

The remaining parts of this paper are organized as follows: Section 2 gives the definition of resources and participants in a network and specifies the different fairness methods. Based on these definitions, the mathematical model is developed in Section 3. Metrics for the performance evaluation are discussed in Section 4, whereas, for the evaluation itself, a number of example scenarios are investigated in Section 5. Finally, Section 6 gives a conclusion and an outlook.

## 2. Terminology

When discussing fairness aspects of multipath transport, some terms and definitions have to be specified to avoid ambiguities. Inside a multipath network, there are *participants* who share a common *resource*. An analogous example is a number of wireless stations, which, as participants, share the wireless medium which is the resource. Resources and participants are defined in Sections 2.1 and 2.2. Furthermore, clarification is needed as to what exactly the notion of *fairness* means, which is covered in Section 2.3.

The discussions in this paper at several occasions refer to (MP)TCP to identify different types of connections such as flows or subflows. The MPTCP-related wording should, however, be interpreted without loss of generality; it can be transferred to any multipath supporting transport protocol.

### 2.1. Resources

The following definitions of the resources are illustrated by the example in Figure 1. The blue circles identify the interfaces, the small grey dots attached to the nodes are the interfaces. The black arrows and numbers denote the links with their capacities in Mbps.

**Definition 1.** *Network: A network is a directed graph specified by*

- *A set of nodes V in the example network in Figure 1—the nodes a to e.*
- *A set of interfaces L that is greater than or equal to the number of nodes in the network. Each node has at least one interface, and each interface can only be part of exactly one node. In Figure 1, L is formed by the interfaces a1, a2, . . . e.*
- *A connectivity matrix with elements $cap_{ij} \geq 0$, $i, j \in L$. If $i, j$ are directly connected by the same link, then $cap_{ij}$ specifies the link capacity in the direction from i to j; otherwise, $cap_{ij} = 0$. If $i, j$ belong to the same node, it is assumed that $cap_{ij} \to \infty$. This means that nodes which act as forwarders, which could in practice be e.g., routers, are assumed to have infinite processing capacity—bottlenecks only occur on the links between nodes, but not inside the nodes. The connectivity matrix for the example of Figure 1 is given in Table 1. Empty elements have a value of 0 that is omitted for better overview.*

The definition of a network raises the question as to why the additional notions of the node and interface set are required for the definitions of the network's directed graph, besides the connectivity matrix. The reason are requirements of the linear programming formulation where interfaces have to be distinguished from the nodes.

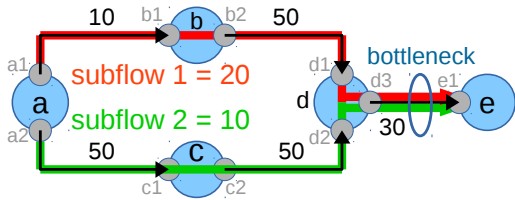

**Figure 1.** Example network to explain Definitions 1 to 5. Blue circles: nodes; grey dots: interfaces; black arrows/numbers: links with capacities; colored arrows/numbers: flows with capacity assignments. All numerical values are given in Mbps.

**Table 1.** Connectivity matrix for example network in Figure 1. Rows: source, columns: destination. All values in Mbps. Zero values are omitted for better overview.

|     | a1 | a2 | b1 | b2 | c1 | c2 | d1 | d2 | d3 | e1 |
|-----|----|----|----|----|----|----|----|----|----|----|
| a1  | ∞  | ∞  | 50 |    |    |    |    |    |    |    |
| a2  | ∞  | ∞  |    |    | 10 |    |    |    |    |    |
| b1  |    |    | ∞  | ∞  |    |    |    |    |    |    |
| b2  |    |    | ∞  | ∞  |    |    | 50 |    |    |    |
| c1  |    |    |    |    | ∞  | ∞  |    |    |    |    |
| c2  |    |    |    |    | ∞  | ∞  |    | 50 |    |    |
| d1  |    |    |    |    |    |    | ∞  | ∞  | ∞  |    |
| d2  |    |    |    |    |    |    | ∞  | ∞  | ∞  |    |
| d3  |    |    |    |    |    |    | ∞  | ∞  | ∞  | 30 |
| e1  |    |    |    |    |    |    |    |    |    | ∞  |

**Definition 2.** *Path: When a data packet is forwarded from the sender to the receiver, it propagates across a number of nodes resp. their interfaces. A path is defined as the sequence of interfaces which the packet passes while being forwarded. The first interface of the path is the one of the sending node, the last one is the receiving node's interface.*

*The bandwidth of a path is the bandwidth of the path's slowest link, i.e., the "weakest link in the chain".*

In the example of Figure 1, node a is the sender and node e the receiver. There are two possible paths $[a, b, d, e]$ and $[a, c, d, e]$ which are identified by the red and green arrows, respectively.

**Definition 3.** *Bottleneck: A link whose capacity is fully utilized by flows or subflows limits the speed of at least one (sub)flow. Such a link is called a bottleneck.*

In Figure 1, the path $[a, b, d, e]$ is bottlenecked by link $(a, b)$ which has a capacity of only 10 Mbps. Path $[a, c, d, e]$ is bottlenecked by link $(d, e)$ which allows a maximum 30 Mbps. Out of this capacity, 10 Mbps are already used by the data transport via path $[a, b, d, e]$ so that only 20 Mbps are left.

*2.2. Participants*

**Definition 4.** *Flow: The set of all data packets belonging to the same communication, e.g., an individual file transfer between two nodes, regardless of the interfaces used at the sender and receiver side and the path(s) between the sender and receiver, is denoted as a flow.*

In Figure 1, all data transported from the sending node "a" to the receiving node "e" is a flow. The set of all traffic demands resp. data flows (single or multipath) is denoted as *K*.

**Definition 5.** *Subflow: A subflow occurs when a flow is split into multiple paths; it denotes the packets of a flow that use a certain interface at the sender and receiver and a certain path inside the network.*

In Figure 1, there are two subflows that are equivalent to the available paths identified in Definition 2.

## 2.3. Fairness

Since TCP traffic is elastic, each participant—a flow or subflow—has a potentially unlimited demand which is restricted by the limited capacity of the links. The task of fairness is to share the link capacity in a proper way between the participants. From the view of a network operator, ensuring no user has to starve is a crucial aspect of maintaining user satisfaction. However, no absolute value for a guaranteed minimum capacity can be specified because it cannot be predicted how many flows will share a link.

For reasons of simplicity and better overview, in the further figures with network examples, the interfaces are omitted and the (sub)flows are drawn besides the links.

**Definition 6.** *Bottleneck subflow fairness (BSF): A bottlenecked link may be shared by a number of multipath TCP subflows along with legacy TCP flows which compete for the available bandwidth. The idea of BSF is that each MPTCP subflow should get the same capacity share like a TCP flow [21].*

Figure 2 shows an example network scenario which illustrates the BSF mechanism. There are six nodes $a, b, c, d, e, m$ and two flows $[a, e]$ and $[m, e]$ inside the network. All link capacities are given in Mbps. Multipath flow $[a, e]$ has two subflows $[a, b, d, e]$ and $[a, c, d, e]$ in this scenario. Both subflows share the same bottleneck link $(d, e)$ with the single path flow $[m, e]$. Thus, for bottleneck subflow, for fair allocation, each subflow should get the same allocation as the single path flow on the bottleneck link $(d, e)$. This corresponds to a capacity allocation of 20 Mbps for flow $[a, e]$ and 10 Mbps for flow $[m, e]$.

Let $sf_i$ where $i = 1, \ldots, n$ be $n$ legacy TCP flows or MPTCP subflows sharing a link $(g, h)$ with capacity $cap_{gh}$. If all $sf_i$ are bottlenecked on $(g, h)$, the assignment $sf\_alloc_i$ for each $sf_i$ is

$$sf\_alloc_i = cap_{gh}/n \quad \text{for } i = 1, \ldots, n. \tag{1}$$

The more common case is that a subset of the (sub)flows $sf_1, \ldots, sf_m$, $m < n$ is already bottlenecked on other links where each of them gets an assignment $sf\_alloc_i < cap_{gh}/n$, $i = 1, \ldots, m$. The remaining $sf_{m+1}, \ldots, sf_n$ which are bottlenecked on $(g, h)$ then get the assignment

$$sf\_alloc_i = \frac{cap_{gh} - \sum_{j=1}^{m} sf\_alloc_j}{n - m} \quad \text{for } i = m+1, \ldots, n. \tag{2}$$

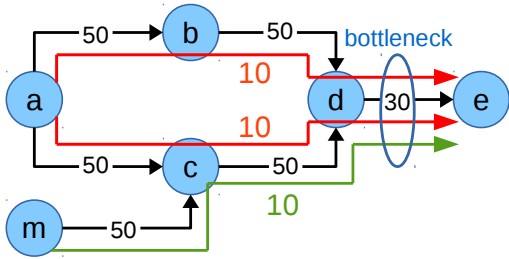

**Figure 2.** Example for bottleneck subflow fair allocation. Blue circles: nodes; black arrows/numbers: links with capacities; colored arrows/numbers: flows with capacity assignments. All numerical values are given in Mbps.

**Definition 7.** *Bottleneck flow fairness (BFF): An MPTCP flow may have more than one subflow on the same shared bottleneck link. The BFF approach requires that all subflows of the same MPTCP flow should get the same*

*aggregated share as a single legacy TCP flow [21]. In other words, an MPTCP user should not get an advantage by deploying multiple subflows on the same link. Practical methods based on coupling the congestion window sizes of individual subflows [22] as well as feedback-based path failure (FPF) or buffer blocking protection (BBP) [23] have been proposed to ensure that an MPTCP flow gets at least the same share as a legacy TCP flow and avoid (sub) flows being underutilized.*

In Figure 3, subflows $[a, c, d, e]$ and $[a, b, d, e]$ of the multipath flow $[a, e]$ are coupled together and considered as a single flow on bottleneck link $(d, e)$. According to the bottleneck flow fair allocation, the bottleneck capacity of link $(d, e)$ should be shared equally between the competing flows $[a, e]$ and $[m, e]$. This corresponds to an allocation of 15 Mbps to each flow.

Let $f_i$ where $i = 1, \dots, n$ be $n$ legacy or multipath flows sharing a link $(g, h)$ with capacity $cap_{gh}$. If all $f_i$ are bottlenecked on $(g, h)$, the assignment $f\_alloc_i$ for each $f_i$ is

$$f\_alloc_i = cap_{gh}/n \quad \text{for } i = 1, \dots, n. \tag{3}$$

The more common case is that a subset of the flows $f_1, \dots, f_m$, $m < n$ are already bottlenecked on other links where each of them gets an assignment $f\_alloc_i < cap_{gh}/n$, $i = 1, \dots, m$. The remaining $f_{m+1}, \dots, f_n$ which are bottlenecked on $(g, h)$ then get the assignment

$$f\_alloc_i = \frac{cap_{gh} - \sum_{j=1}^{m} f\_alloc_j}{n - m} \quad \text{for } i = m + 1, \dots, n. \tag{4}$$

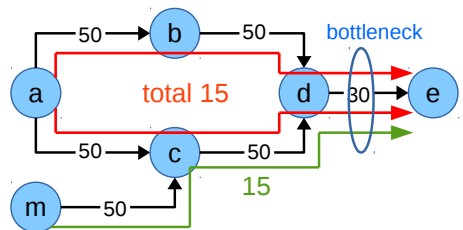

**Figure 3.** Example for bottleneck flow fair allocation. Blue circles: nodes; black arrows/numbers: links with capacities; colored arrows/numbers: flows with capacity assignments. All numerical values in Mbps.

**Definition 8.** *Network flow fairness (NFF):*

*In contrast to the two previously mentioned fairness methods where the bottleneck is the shared resource, for NFF, it is the entire network. If an MPTCP flow has multiple subflows, each of these subflows may propagate along a different path and thus become bottlenecked on different links and experience different amounts of congestion. The aggregated throughput of a flow is the sum of the throughputs of all subflows belonging to that particular flow. To overcome the problem of unequal share between different flows, a resource pooling (RP) algorithm has been defined which aims at balancing the amount of congestion which the different (sub)flows have to face [24]. If a link is heavily congested, the algorithm tries to find less congested alternative paths for some of the (sub)flows sharing that link so that some of the load can be removed from the link. In other words, the available links in the whole network are considered as a pool of resources which should be shared in a fair way between the participants. The benefits of balancing the load between participants using resource pooling based congestion control are decreased overall network congestion, increased efficiency and reliability [25]. This fairness mechanism is called Network Flow Fair (NFF) allocation. Based on the resource pooling principle, several congestion control algorithms have been proposed such as Linked Increases Algorithm (LIA) [26], Opportunistic Linked Increases Algorithm (OLIA) [7], Adapted OLIA [27] or Balanced link adaptation (Balia) [8] for MPTCP and Resource Pooling Multipath version 2 (RP-MPv2) for CMT-SCTP [28]. Several MPTCP algorithms have been investigated analytically concerning their TCP friendliness and stability [8].*

For an NFF example, consider the network in Figure 4 which includes eight nodes a, b, c, d, e, f, m and n. The link capacities are given in Mbps. There are two flows [a, f] and [m, n] inside the network. Flow [m, n] is limited by link (m, b) and flow [a, f] is limited by the links (b, c) and (d, e). Thus, link (m, b), (b, c) and (d, e) are the bottleneck links for this scenario. Flow [m, n] can not get more than 10 Mbps which makes the ideal fair share for flow [a, f]. However, allocating 10 Mbps to both flows would not utilize the network capacity fully. After being fair, for efficient network utilization flow, [a, f] can get extra 10 Mbps from the network ("fair+spare"). For this scenario, network flow fair allocates [a, f] to 20 Mbps and [m, n] to 10 Mbps.

Due to the complexity of interdependence between flows in a network-wide view, it is hard to give a closed expression for a fair capacity assignment as it was done for BSF and BFF. It is however possible to express NFF in the MINLP model as specified in the paragraph about NFF in Section 3.2.

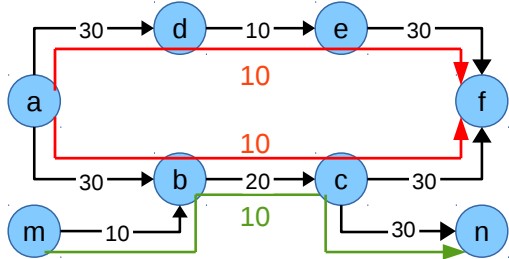

**Figure 4.** Example for network flow fair allocation. Blue circles: nodes; black arrows/numbers: links with capacities. Colored arrows/numbers: flows with capacity assignments. All numerical values in Mbps.

## 2.4. Efficiency vs. Fairness

The performance of sending data across a network can be characterized by efficiency or fairness. There are two ways to measure *efficiency*: On the one hand, the focus can be on the individual links inside a network, which is the method which is preferred in the literature e.g., [29]. What is the load on each link in relation to the capacity of the respective link is observed. On the other hand, the entire network can be under consideration: when summing up the throughput of all flows which are transported by the network, the result is the aggregated capacity of the network. A high link utilization does not necessarily result in a high aggregated capacity, e.g., if because of bad network design a large number of links become bottlenecks, the link utilization is high, whereas the aggregated capacity may still be low since the transported traffic is slowed down due to these bottlenecks.

In opposition to a pure view on maximizing the network utilization, one can also aim at providing maximum *fairness* between the allocation of capacity to a number of flows, which means in the best case that each flow gets the same amount of capacity.

In many cases, it is not possible to maximize both efficiency and fairness, i.e., they form a trade-off, so that a network operator has to find a compromise between both metrics. The scenario depicted in in Figure 5 shows the relationship between network utilization and fairness. Three nodes a, b and c are connected by two links (a, b) and (b, c) which have the same capacity of 100 Mbps. There are two data flows [a, c] and [b, c]. The link (a, b) is only occupied by flow [a, c] whereas link (b, c) is used by both flows, which results in a bottleneck situation. If the aim is optimum resource utilization of the network, flow [a, c] will be allocated the entire capacities of both links which results in full link utilization, but minimum fairness because flow [b, c] does not achieve any throughput. If the focus is on fairness, flows [a, c] and [b, c] share the bottlenecked link (b, c) by 50 Mbps each, which, however, results in the problem that link (a, b) is now only utilized by 50 Mbps. The given scenario is an example where fairness only can be achieved at the expense of network utilization. Weighting can be applied to specify by what extent efficiency or fairness should be considered when assigning resources. In the discussed example, a factor $\alpha$ is set to 0 if utilization should be the only goal and fairness should not

be considered at all, whereras it is set to 1 if fairness should be pursued irrespective of any efficiency loss. Figure 6 shows the amount of average link utilization dependent on $\alpha$ for the scenario in Figure 5. In case of $\alpha = 0$, both links are fully utilized, so the average utilization is 1. In case of $\alpha = 1$, link $(a, b)$ has a utilization of 0.5 and link $(b, c)$ is fully utilized, so the average utilization is 0.75.

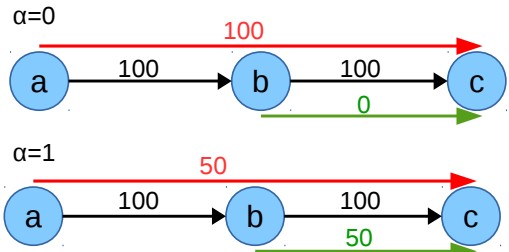

**Figure 5.** Efficiency ($\alpha = 0$) vs. fairness ($\alpha = 1$) example. Blue circles: nodes; black arrows: links; colored arrows: data flows; numerical values: link speeds resp. flow assignments in Mbps.

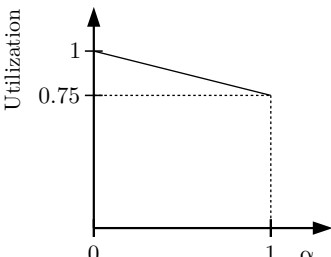

**Figure 6.** Efficiency-fairness graph for scenario in Figure 5, based on Ref. [30].

## 3. Mathematical Model

The aim of this paper is the introduction of a novel optimum mathematical method to deliver data (sub)flows across a network considering the fairness-efficiency trade-off that was already mentioned in Section 2. High efficiency means that the aggregate throughput of all (sub)flows should be maximized by utilizing the available links to the maximum possible extent. Fairness means that no (sub)flow should starve by assigning only a small amount of link capacity. Both goals cannot be achieved at the same time; a compromise has to be identified as discussed in Section 2.4.

The resource allocation of (MP)TCP flows to bandwidth-limited links leads to a mixed-integer nonlinear programming (MINLP) problem. In general, an (N)LP problem is defined by an objective function which takes a number of input arguments that are limited by constraints. The task of the (N)LP solver is to find values for the input arguments which yield the maximum value for the objective function while keeping the constraints. The MINLP model is developed in two steps. In Section 3.1, the basic model is shown in case throughput is the only value to be optimized. In Section 3.2, the objective function of the model is extended for the different fairness approaches.

### 3.1. MINLP Model without Fairness

The objective function, in case no fairness is considered, only specifies the aggregate throughput of all flows in the network. It does make use of multipath transport for a given flow, but only if a higher aggregate throughput for the entire network can be achieved. Let $K$ be the set of all traffic demands' resp. data flows in the network. Let $[s, t] \in K$ be a flow between the stations $s$ and $t$ and $\varphi_{st}$ be the capacity which is allocated to that flow. The symbol $\varphi_{st}$ which denotes, as already said, the *assigned* capacity to a single or multipath flow should not be confused with the *fixed* capacity $cap_{gh}$ of a link; they are not necessarily identical. Let *Alloc* be the aggregated allocation of $K$ as shown in Equation (5):

*Aggregated allocation of all flows*:

$$Alloc = \sum_{[s,t] \in K} \varphi^{st}. \tag{5}$$

In case that only the aggregated allocation should be maximized and the fairness is not considered, which is called max-flow allocation, the objective function then only consists of the variable *Alloc* as given in Equation (6):

*Max-flow allocation objective:*

$$\text{maximize}: \quad Alloc. \tag{6}$$

The constraints for the max-flow model apply to any network with single or multipath data flows. They are also used in the fairness implementations described later where they are supplemented by fairness-specific constraints. In this section, the constraints are discussed in a textual way, the formal description is given in Appendix A. Specifically, the formulations for the general constraints independent from the allocation method are given in the Appendix A.1. The equation numbers in parentheses starting with "A" refer to the Appendix A.

*Flow conservation constraint* (Ref. [16]): At each node except for the communication endpoints, the number of ingoing and outgoing flows must be equal; routers do not act as source or sink (Equation (A1)).

*Capacity limitation constraints* (Ref. [16]):

- The overall throughput on a link, summed up for all flows, cannot be higher than the physical speed of the link (Equation (A2)).
- A flow can only occupy capacity on a link if the latter is part of the flow's path. The maximum available capcity is the physical speed of the link (Equation (A3)).

*Path constraint*: In a given network, there are interfaces which act as a data source while others become a data sink. The number of available paths cannot exceed the product of sources and sinks, i.e., the data between each pair of connected interfaces is transferred via exactly one path (Equation (A4)). Due to the fact that a subflow corresponds to exactly one path, the same constraint also holds for the number of subflows (Equation (A5)).

*Multipath flow identifier constraint*: In the (N)LP formulation, variables have to be specified to identify whether or not a node is part of any path between a source and a sink node, i.e., acting as an intermediate node (Equations (A6) and (A7)). Further variables identify whether a flow which makes use of a node is a multipath flow (Equations (A8) and (A9)).

*Multipath subflow identification constraints*: These constraints control a set of helper variables, where one variable exists for each combination of any subflow and any link. By means of these variables, the following constraints are enforced:

- A multipath flow which does not use a particular link cannot have a subflow on that link (Equation (A10)).
- If a link is used by a multipath flow, at least one of the subflows has to use that link (Equation (A11)).
- A given subflow can occur at maximum once on a particular link (Equation (A12)).
- Keep track of whether a flow has a subflow on a particular link (Equation (A13)).
- Keep track of whether a flow divides into subflows at the source node (Equation (A14)).
- A multipath-enabled flow may not have a subflow yet because none might have been computed yet during the solution process. After the computation, the flow may be assigned one or more than one subflow (Equation (A15)).
- A data flow cannot be split into subflows if both end nodes only have one interface, respectively (Equation (A16)).

The previously mentioned equations concerning subflows ensure that the creation of subflows is logically correct, but do not limit the number of subflows. This is ensured by a second equation set:

- Keep track of how many subflows a flow is split into (Equation (A17)).
- The number of subflows between a sender and receiver node cannot be larger than the product of sender and receiver interfaces (Equation (A18)).
- The number of subflows is equivalent to the number of paths between the sender and the receiver node (Equations (A19) to (A21)).
- No subflow can be created if both the sender and the receiver only have one interface. Between a pair of sender and receiver interfaces, at a maximum, one subflow can be created (Equations (A22) to (A24)).

*Congested links' constraints for flows*: The aim of transport protocols is to maximize link utilization in order to make efficient usage of the network. This means that each flow should experience at least one congested link; if all links which are occupied by a flow had capacity left, it would mean that there is space left for additional throughput. The corresponding equations express the following constraints:

- There is at least one congested link for each flow (Equation (A25)).
- A congested link is fully utilized by flows, there is no capacity left (Equation (A26)).
- A link can be on a path for a flow even though it might not be fully utilized and the bottleneck link for a subflow must be on the path allocated to that subflow (Equation (A27)).

*Congested link constraints for subflows*: If a flow splits into multiple subflows, each of them should be a part of at least one congested link, as it is the case for single-path flows. It can however happen that on a given link two or more subflows belong to the same flow. The following constraints describe links congested by subflows:

- A necessary condition that a link is identified as congested by a particular subflow is that the subflow exists on that link (Equation (A28)).
- Each subflow has at least one congested link (Equation (A29)).
- A subflow cannot be congested on a link if the parent flow is not congested on that link (Equation (A30)).
- If a flow is bottlenecked on a link, then all its subflows are bottlenecked on that link (Equation (A31)).

*3.2. Extension of the Objective Function for Fairness*

The objective function given in Equation (5) does not consider fairness but is only targeted at maximizing the aggregate throughput. In this section, the objective function is extended for the different fairness methods BSF, BFF and NFF. The formulations for the constraints specific for the different allocation methods are given in the Appendix A.2.

3.2.1. Bottleneck Subflow Fair Allocation (BSF)

BSF ensures equal share between multiple subflows on a common congested link as described in Definition 6. An example for a network where BSF is applied was already given in Figure 2.

In order to formulate bottleneck subflow fair allocation, let $m^{st}$ be the end-to-end capacity allocation to a *multipath* flow $[s, t] \in K$, $m^{st} = 0$ if the flow $[s, t] \in K$ is a single path flow. The value $m^{st}$ should not be confused with $\varphi^{st}$ which is the allocation to *any* flow, irrespective of single or multipath, or with $cap_{ij}$ which is the physical capacity of a link.

In the most simple case, the objective function for BSF is the same as the previously defined max-flow Equation (6) which does not make use of multiple paths if a single-path solution yields a better result. The objective function is therefore written as:

*BSF-I:*

$$\text{maximize}: \quad Alloc. \tag{7}$$

The variable *Alloc* is the aggregated allocation of all flows (multipath + single path).

The difference between pure max-flow and BSF-I with maximum allocation is that BSF includes additional constraints enforcing fairness which are applied in addition to the general constraints from Section 3.1:

- All subflows which share the same bottlenecked link should get the same share (Equations (A40) and (A41)).
- It has to be ensured that a single-path flow gets the same share as a subflow of a multipath flow (Equations (A42) and (A43)).

BSF-I supports multipath, but does not make use of it if single-path yields a better allocation. The goal is, however, to push the system towards multipath usage which means that the latter should be rewarded to enhance fairness. Let $M$ be the aggregated allocation of all multipath flows as shown in Equation (8)

$$M = \sum_{[s,t]\in K} m^{st}, \tag{8}$$

where $s$ and $t$ are stations inside the network $K$ running multipath flows. The objective function is then extended as:

*BSF-II:*

$$\text{maximize}: \quad Alloc + \beta \cdot M, \tag{9}$$

where $\beta$ is a positive constant to provide a weighting between prioritizing maximum capacity and using multiple paths. The elements $m^{st}$ which are summed up to $M$ are determined by constraints given in the Appendix A.2.1 in Equations (A32) to (A34). Further constraints control the mapping between flows, subflows and links:

- A subflow only gets an allocation on a particular link if the link is part of the particular subflow (Equations (A35) and (A36))
- The allocation of a flow is greater than or equal to the allocation of its subflows on a given link (Equations (A37) and (A38)).
- A subflow gets the same allocation on all links which are part of the subflow's path (Equation (A39)).

Finally, for BSF-II, the constraints controlling the fairness as mentioned for BSF-I also need to be applied.

Practical multipath implementations as they are e.g., known from MPTCP always make use of multiple paths if the end nodes are equipped with multiple interfaces, even if a single-path solution might yield a higher aggregated throughput. This happens e.g., if a subflow shares more than one bottleneck link with other (sub)flows. Therefore, BSF-II reflects the behavior of practical protocols in a better way than BSF-I.

### 3.2.2. Bottleneck Flow Fair Allocation (BFF)

The Bottleneck Flow Fair (BFF) allocation method assigns the same capacity to all flows on a shared bottleneck link, irrespective of the fact if more the flow occupies the link with more than one subflows, which was explained in Definition 7. In other words, BFF ensures that there is no advantage for multipath flows over single path flows competing for the same link. An example for a network where BFF is applied was already given in Figure 3.

As it was shown for BSF, there are two ways to specify the objective function for BFF, namely maximizing the aggregated allocation or supporting a high amount of flows using multipath transport. This results in the objective functions BFF-I (Equation (10)) and BFF-II (Equation (11)) which are identical to the ones specified for BSF:

*BFF-I*

$$\text{maximize}: \quad Alloc, \tag{10}$$

*BFF-II*

$$\text{maximize}: \quad Alloc + \beta \cdot M, \tag{11}$$

where *Alloc*, *β* and *M* are defined in the same way as for BSF.

The constraints which have to be applied specifically for BFF are:

- All flows on a shared bottleneck link should get the same allocation (Equations (A44) and (A45)).
- Optionally, all subflows of a multipath flow which share the same bottleneck link are assigned the same allocation (Equations (A46) and (A47)).

After performing the aforementioned operation, BFF optionally takes a second step if there are flows which are represented by more than one subflow on a particular link. For each of these flows, BFF tries to share the capacity assigned to the respective flow equally among the subflows on that link.

### 3.2.3. Network Flow Fair Allocation (NFF)

In Network Flow Fairness, the entire network is the resource whose capacity should be equally shared between flows. This network capacity is determined by summing up the throughput of all flows inside the network, or their subflows in case of multipath transport. Unlike the link capacity which can be easily specified due to the physical nature of a link, the network capacity is difficult to calculate as already mentioned in Definition 8. It depends on the topology of the network, the location of source and sink nodes and which flows or subflows get bottlenecked on which link. Furthermore, there is a trade-off between the goal of resource usage maximization and fairness as discussed in Section 2.3: using multipath might on the one hand be desired to maximize network utilization but on the other hand be unwanted for fairness reasons to avoid that a flow occupies multiple paths which should be assigned to other flows. An example for a network where NFF is applied was already given in Figure 4.

When the entire network is the resource, fairness cannot be expressed by link-level constraints in the LP formulation but can only be included in the objective function. It is proposed to reflect the network fairness by a negative term whose absolute value increases in case of a large difference between the capacity assignments among the flows.

The objective function for NFF is developed in multiple steps. After each step, an example of a network architecture is shown where the fairness methods specified up to that step fail to ensure fair allocation of the flows and thus require an extension which is then shown in the next step; the respective objective functions are denoted as NFF-objective-I, -II, etc.

Let $flow\_diff^{st}$ be the sum of the allocation differences between flow $[s, t] \in K$ and all other flows $[q, r] \in K$. $\delta_{\text{all}}$ estimates the total allocation differences between all flows not concerning whether the flows share a common congested link or not. In Equation (13), the denominator is introduced because the allocation difference between two flows is added twice, between flows $s$, $t$ and vice versa:

$$flow\_diff^{st} = \sum_{(q,r) \in K} | (m^{st} - m^{qr}) | \qquad \forall [s,t] \in K, \tag{12}$$

$$\delta_{\text{all}} = \frac{\sum_{[s,t] \in K} flow\_diff^{st}}{2}. \tag{13}$$

The objective function NFF–objective-I (Equation (14)) maximizes the differences of the aggregated allocation (*Alloc*) and the total allocation difference among the flows ($\delta_{\text{all}}$). Though it seems that the objective function provides a network flow fair solution, it does not always fully utilize the available network capacity, as shown in the previously discussed scenario in Figure 4. As explained earlier, according to the network flow fair allocation method, flow $[a, f]$ should get 20 Mbps and flow $[m, n]$ should get 10 Mbps from the network. This implies the objective function value of 20 ($Alloc = 20 + 10$, $\delta_{\text{all}} = 20 - 10$) for this scenario. However, if each flow gets 10 Mbps, then the optimum value from the the objective function remains the same ($10 + 10 - 0 = 20$) as there is no

difference between the flow allocations. Thus, both allocations are valid and optimum for this scenario; however, they differ in the fairness:

*NFF-objective-I:*

$$\text{maximize}: \quad Alloc - \delta_{\text{all}}. \tag{14}$$

Nevertheless, if the solver allocates 10 Mbps to each flow, the network capacity is not fully utilized, which is not desirable. From the closer look, it is visible that the flow [a, f] does not utilize multipath though the flow is multipath capable and not affecting any other flow. Based on these insights, the objective function is extended to Equation (15) in order to push the system towards multipath if flows are multipath capable:

*NFF-objective-II:*

$$\text{maximize}: \quad Alloc - \delta_{\text{all}} + M. \tag{15}$$

$M$ is the aggregated allocation of all multipath flows as defined in Equation (8). Though the objective function solves the above-mentioned problem for the scenario in Figure 4, adding $M$ to the objective function might not be enough for ideal network flow fair allocation. Consider the network scenario in Figure 7. The network is composed of eight nodes a, b, c, d, e, f, m and n. All link capacities are in Mbps. There are five flows [e, a], [a, d], [b, f], [m, c] and [d, n] inside the network. The ideal network flow fair solution would be flow [e, a] = 10 Mbps, [a, d] = 110 Mbps, [b, f] = 10 Mbps, [m, c] = 10 Mbps and [d, n] = 10 Mbps. Now, the value of the objective function NFF-objective-II can be calculated as follows:

$Alloc = 10 + 110 + 10 + 10 + 10 = 150,$
$\delta_{all} = (110 - 10) + (110 - 10) + (110 - 10) + (110 - 10) = 400,$
$M = 110,$
NFF-objective-II $= 150 - 400 + 110 = -140.$

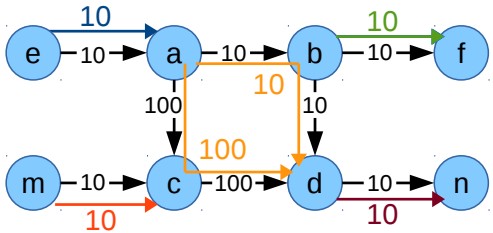

**Figure 7.** Example scenario for Network Flow Fair (NFF) with optimum allocation. Blue circles: nodes; black arrows/numbers: links with capacities; colored arrows/numbers: flows with capacity assignments. All numerical values in Mbps.

The allocations of the flows are [e, a] = 10 Mbps, [a, d] = 10 Mbps, [b, f] = 10 Mbps, [m, c] = 10 Mbps and [d, n] = 10 Mbps where multipath capable flow [a, d] does not utilize multipath then the value of the objective function NFF-objective-II is:

$Alloc = 10 + 10 + 10 + 10 + 10 = 50,$
$\delta_{\text{all}} = 0,$
$M = 0,$
NFF-objective-II $= 50.$

This means the latter allocation where flow [a, d] does not utilize multipath is the optimum solution of the objective function NFF-objective-II for this scenario. If flow [a, d] was assigned to the 100 Mbps path via node c in addition to the 10 Mbps path via node b, there would be a high allocation

difference between flow $[a, d]$ and each of the other flows. The penalization of this high difference by the negative $\delta_{\text{all}}$ element in the objective function outweighs the multipath capacity gain $M$.

In order to tackle this problem, a multiplication factor $|K| - 1$ for the capacity gain is introduced, where $|K|$ is the number of flows inside flow set $K$ and $M_{\text{gain}}$ is the sum of the overall allocation gain due to multipath. The modified objective function is given in Equation (16):

*NFF-objective-III:*

$$\text{maximize}: \quad Alloc - \delta_{\text{all}} + (|K| - 1)M_{\text{gain}}. \tag{16}$$

To formulate the equation sets for calculating $M_{\text{gain}}$, let $max\_sf\_val^{st}$ be the maximum subflow allocation for the multipath flow $[s, t] \in K$. If the flow $[s, t] \in K$ is a single path flow, then $max\_sf\_val^{st} = 0$. $max\_sf\_val^{st}$ is the equivalent single path allocation for the multipath flow $[s, t] \in K$ because, in single path allocation, a flow tries to take the path from which it can get maximum capacity:

$$max\_sf\_val^{st} = \max_{(o,h)\in K, (i,j)\in K} sf\_alloc^{st}_{ohij} \quad \forall [s, t] \in K. \tag{17}$$

The variable $sf\_alloc^{st}_{ohij}$ specifies the capacity allocation to the subflow between interface $o$ to $h$, where the subflow is part of the flow from $s$ to $t$ and uses the link between nodes $i$ and $j$.

The overall allocation gain due to multipath $M_{\text{gain}}$ is the difference between the aggregated allocation when flows inside the network utilize multipath and when flows would use single path only. Let, $Alloc_{\text{single-path-flow}}$ be the aggregated allocation when each flow would use single path.

$$Alloc_{\text{single-path-flow}} = \sum_{[s,t]\in K} \varphi^{st}_{\text{single-path}} + \sum_{[s,t]\in K} max\_sf\_val^{st}, \tag{18}$$

where $\varphi^{st}_{\text{single-path}}$ is the allocation of the single path flow $(s, t) \in K$ which is computed by the constraints given in Equations (A48) to (A50) in the Appendix A.2.3.

The overall allocation gain due to multipath is then:

$$M_{\text{gain}} = Alloc - Alloc_{\text{single-path-flow}}. \tag{19}$$

Though the objective function NFF-objective-III (Equation (16)) overcomes the limitation of the objective function NFF-objective-II for the scenario in Figure 8, it may not always provide the optimum allocation. For example, consider the scenario in Figure 9. The allocation of the flow $[m, n]$ is limited by the links $(c, d)$ and $(g, h)$. Flow $[a, d]$ is limited by the link $(a, b)$ and the flow $[e, h]$ is limited by the link $(g, h)$. This leads to the ideal network flow fair allocation of 20 Mbps to each flow by utilizing the whole network capacity. The corresponding value of the objective function NFF-objective-III $= (20 + 20 + 20) - 0 + (3 - 1) \times 0 = 60$. However, if the allocation is considered as in Figure 10 where the allocation of the flows $[a, f]$, $[e, h]$ and $[m, n]$ are 30, 30 and 10 Mbps, respectively, flows $[a, f]$ and $[e, h]$ limit the allocation of the flow $[m, n]$ by utilizing multipath. In this case, the multipath gain of each individual multipath flow is 10 Mbps. This results in a value of the objective function NFF-objective-III $= (30 + 30 + 10) - ((30 - 30) + (30 - 10) + (30 - 10)) + (3 - 1) \times (10 + 10) = 70$, which is higher than the previous allocation. Thus, the objective function NFF-objective-III would allocate 30 Mbps to the flows $[a, f]$ and $[e, h]$, whereas flow $[m, n]$ would be allocated 10 Mbps, which is not an optimum network flow fair solution.

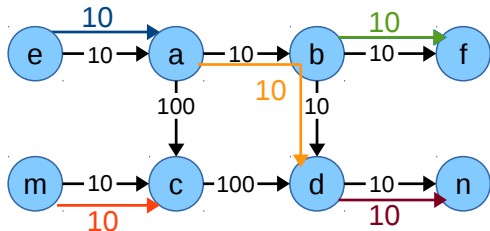

**Figure 8.** Scenario as in Figure 7, but NFF-objective-II prevents flow [a, d] from using multipath although it does not harm any other flow.

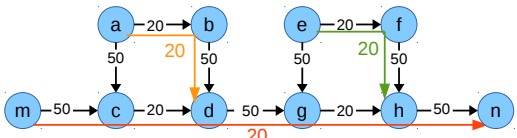

**Figure 9.** Example scenario for NFF with optimum allocation. Blue circles: nodes; black arrows/ numbers: links with capacities; colored arrows/numbers: flows with capacity assignments. All values in Mbps.

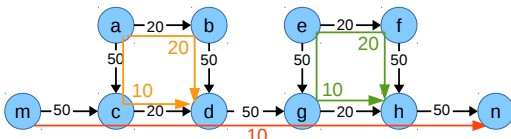

**Figure 10.** Scenario from Figure 9, but NFF-objective-III enforces multipath for flows [a, d] and [e, h] and as a result restricts flow [m, n].

Going back to the objective function NFF-objective-II, the reason behind the problem in Figure 8 was comparing all flows for the network flow fair solution even if they do not share any congested links, which means they are disjoint. Comparing all flows might create higher allocation differences which can not be compensated by the $M$ value. Introducing $M_{\text{gain}}$ with the multiplication factor $\beta$ is more biased to multipath allocation, which has a negative effect on fairness. This results in the idea of comparing flows only when they share a congested link.

To compare flows according to the shared congested link, let $cong\_flow\_diff_{ij}^{st}$ be the sum of the allocation differences between the flow $[s, t] \in K$ and all other flows sharing the congested link $(i, j) \in A$ with the flow $[s, t] \in K$. In addition, consider a binary variable $cong\_group\_id_{ij}^{st}$ which identifies the flow $[s, t] \in K$ sharing the congested link $(i, j) \in A$. $cong\_group\_id_{ij}^{st} = 1$ when the flow $[s, t] \in K$ shares the congested link $(i, j) \in A$ with other flows. Equation (20) means when flow $[s, t] \in K$ and all other flows $[q, r] \in K$ sharing the same congested link $(i, j) \in A$ i.e., $cong\_group\_id_{ij}^{st} = 1$ and $cong\_group\_id_{ij}^{qr} = 1$ then $cong\_flow\_diff_{ij}^{st}$ is the sum of the allocation differences between the flow $[s, t] \in K$ and all other flows $(q, r) \in K$. After calculating the allocation differences between the same congested flow group, $\delta_{\text{congested}}$ sums the total allocation difference of all those flows:

$$
\begin{aligned}
cong\_flow\_diff_{ij}^{st} \\
= \sum_{(q,r) \in K} | \, (cong\_group\_id_{ij}^{st} \cdot (\varphi^{st} - \varphi^{qr}) \\
\cdot cong\_group\_id_{ij}^{qr}) \, | \\
\forall (i, j) \in K, [s, t] \in K,
\end{aligned}
\tag{20}
$$

$$
\delta_{\text{congested}} = \sum_{[s,t] \in K} \sum_{(i,j) \in A} cong\_flow\_diff_{ij}^{st}.
\tag{21}
$$

Formulating the equation sets to identify the flows sharing the same congested link requires that all links which are congested for the flows $[s, t] \in K$ are identified. A link is said to be congested only when the link is fully utilized. In the constraints, a number of variables is defined which keeps track of whether a link is fully loaded, how many congested flows are on the link and whether these flows share the link with flows which are congested on another link.

Equation (22) is the revised objective function for network flow fair allocation comparing the flows sharing the same congested link. $\delta_{\text{congested}}$ is dependent on the allocation of flows sharing a congested link which might lead to having a smaller number of common congested links among the flows which means multipath capable flows might not use multipath in cases where using multipath would create a common congested link with other flows. Consider the scenario in Figure 11 with two flows $[a, f]$ and $[m, n]$. Allocation of the multipath capable flow $[a, f]$ is limited on link $(a, b)$ and $(d, e)$, the single path flow $[m, n]$ is limited on the link $(m, b)$. The resulting maximum allocation is 20 Mbps for flow $[a, f]$ and 90 Mbps for the flow $[m, n]$. This leads to the network flow fair solution of 20 Mbps to the flow $[a, f]$ and 90 Mbps to the flow $[m, n]$ sharing the congested link $(b, c)$. Since the flows share a congested link, their allocation difference is considered in the variable $\delta_{\text{congested}}$ of the objective function NFF-objective-IV. $M$ is the total allocation to multipath flows as defined in Equation (8). The value of the objective function for this allocation is $(20 + 90) - (90 - 20) + 20 = 60$:

*NFF-objective-IV:*

$$\text{maximize}: \quad Alloc - \delta_{\text{congested}} + M. \tag{22}$$

The value of $\delta_{\text{congested}}$ is determined by a number of equations which perform the following tasks:

- Compute the allocation on a link or if it is congested (Equations (A51) and (A52)).
- Compute how many flows on a link are congested (Equations (A53) to (A55)).
- Check whether a given flow shares a link with other flows (Equations (A56) to (A58)).

However, if the flow $[a, f]$ uses only one path as in Figure 12 and does not use the path $[a, b, c, f]$, then flows $[a, f]$ and $[m, n]$ do not have a shared congested link which would nullify the value of $\delta_{\text{congested}}$ and $M$. In that case, the allocations of the flows are $[a, f] = 10$ Mbps and $[m, n] = 90$ Mbps. This implies the value of the objective function NFF-objective-IV is $(10 + 90) - 0 + 0 = 100$, which is higher than the previous allocation. The optimum allocation is then 10 Mbps to the flow $[a, f]$ and 90 Mbps to the flow $[m, n]$ by the objective function NFF-objective-IV, which is not an ideal network flow fair solution for this scenario.

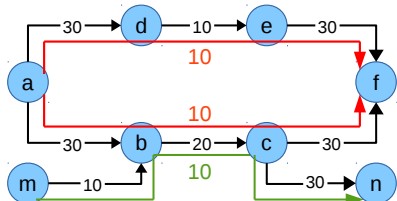

**Figure 11.** Example scenario for NFF with optimum allocation. Black arrows/numbers: links with capacities, colored arrows/numbers: flows with capacity assignments. All values in Mbps.

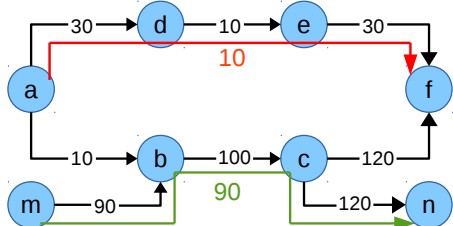

**Figure 12.** Scenario as in Figure 11, but NFF-objective-IV prevents flow $[a, f]$ from using multipath and assigning link $[b, c]$ to flow $[m, n]$ resulting in poor fairness.

All versions of objective functions discussed until now may still insufficiently utilize multipath for some scenarios: multipath may be deployed although it degrades fairness or it may not be deployed despite the requirement of efficient network usage. Therefore, the goal is to identify an objective function which does not affect fairness while maintaining multipath so that it can achieve optimum results for any scenario. In order to specify such objective function, a counter *SF* for the total number of subflows assigned to all flows is introduced:

$$SF = \sum_{[s,t]\in K} n\_sf^{st}. \tag{23}$$

The objective function for the optimum network flow fair solution is formulated in Equation (24) where $\beta$ is the constant multiplication factor which is used to ensure multipath. The value of $\beta$ must be larger than the total network capacity to eliminate the effect of network utilization (*Alloc*) and fairness ($\delta$) on multipath uses. Moreover, *SF* has a negligible adverse effect on fairness because *SF* only specifies the *number* of subflows and does not have any effect on the amount of subflow allocation, which means fairness can be adjusted by providing a very small allocation (as small as 0.01 unit) to unnecessary subflows. On the other hand, by using all the possible paths for multipath flows, total network capacity utilization is ensured by the congested link identifier constraints described earlier in this section:

*NFF-objective-V:*
$$\text{maximize}: \quad Alloc - \delta_{\text{congested}} + \beta \cdot SF. \tag{24}$$

However, NFF-objective-V still might not yield an optimum result if a subflow is assigned a path with a link not used by other flows and that link contributes a smaller capacity to the subflow than a bottlenecked, but a faster link on a different path. The solver would select the unused link to keep $\delta_{\text{congested}}$ low while, on the other hand, the *Alloc* gain would suffer. This suggests that achieving a network flow fair solution which works for any type of scenario is improbable with a single objective function.

In order to cope with the conflict of considering both network allocation and fairness in the objective function in any kind of scenario, the optimization needs to be performed in two steps. In both of them, network utilization as well as fairness are optimized; however, the first step puts the priority on fair capacity assignment, whereas, in the second step, a high network utilization is preferred.

Equations (25) and (26) show the final objective function pair NFF-objective-VI of the two step process, which yields the optimum network flow fair solution. In the first step, a minimum allocation is set for each flow however still utilizing the network capacity as much as possible. The flow allocation of step-1 is used as the minimum allocation constraint for the step-2 to ensure fairness in step-2. $min\_alloc^{st}$ is the allocation of the flow $[s,t] \in K$ in step-1. Equation (27) makes sure that each flow gets the minimum fair share from the network in step-2. If any network capacity has not been allocated, which might affect fair allocation, step-2 makes sure to utilize the spare capacity by allocating fairly among the competing flows. In Equation (26), $|K|$ is the total number of flows inside the network. The $|K|$ multiplication factor helps to prioritize the network utilization (*Alloc*) over fairness ($\delta_{\text{congested}}$) in step-2.

*NFF-objective-VI*
Step 1:
$$\text{maximize}: \quad Alloc - \delta_{\text{congested}} + \beta \cdot SF, \tag{25}$$

Step 2:
$$\text{maximize}: \quad |K| \cdot Alloc - \delta_{\text{congested}}, \tag{26}$$

Minimum allocation constraint for step-2:

$$\varphi^{st} \geq min\_alloc^{st} \qquad \forall [s,t] \in K. \tag{27}$$

## 4. Performance Metrics

The methods to assign link capacity resources in the network to flows discussed in the previous section are evaluated using metrics describing the amount of allocation, the fairness and combinations of both. Definitions of these metrics are given in this section.

### 4.1. Aggregate Allocation

The aggregate allocation which is a metric for the network utilization efficiency is the sum of the throughputs achieved by all (sub)flows in the network:

$$Alloc = \sum_{[s,t] \in K} \varphi^{st}. \tag{28}$$

### 4.2. Jain's Fairness Index ($\gamma_{Jain}$)

In the literature, indexes were developed in order to assess the fairness of resource assignment in networks. A well-known fairness index was proposed by Jain [31], which yields a positive real value, which is $k/n$ in case $k$ out of $n$ users get an equal allocation, whereas the other $n - k$ users are not assigned any capacity. The minimum is $1/n$ if only one user out of $n$ gets an assignment at all, the maximum is 1 in the case all users get the same assignment:

$$\gamma_{Jain} = \frac{(\sum_{[s,t] \in K} \varphi^{st})^2}{|K| \cdot \sum_{[s,t] \in K} (\varphi^{st})^2}. \tag{29}$$

### 4.3. Vardalis's Fairness Index ($\gamma_{Vardalis}$)

This fairness index has been introduced in [32]. It ranges between 0 to 1 for any number of flows. The index is 1 when all flows are receiving equal allocation. If out of $n$ flows, $k$ are equally sharing all the bandwidth and the rest $n - k$ are idle, the fairness index is $(k-1)/(n-1)$:

$$\gamma_{Vardalis} = 1 - \frac{\sum_{[s,t] \in K} |\varphi^{st} - Avg|}{2(|K| - 1) Avg}. \tag{30}$$

The numerator of the fraction in Equation (30) is the sum of the absolute value of the difference of each flows' allocation from the average allocation. The system is more unfair if the sum is more. There are differences between Jain's fairness index (Equation (29)) and Vardalis's fairness index:

- Jain's fairness index ranges from $1/n$ to 1, whereas Vardalis's fairness index ranges from 0 to 1 for $n$ being the number of flows. For example, if there are two flows, one with a certain allocation and other with zero allocation, then Jain's fairness index will be 0.5. On the other hand, the value of the fairness index-II will be 0.
- Vardalis's fairness index is more sensitive to changes, especially when the number of flows is small. For instance, in case of two flows where one flow achieves 50% more bandwidth than the other, Jain's fairness index will be 0.96 where Vardalis's fairness index will be 0.8. When one flow receives exactly twice as much bandwidth as the other, Jain's fairness index will be 0.9 while Vardalis's fairness index will be 0.66.

### 4.4. Normalized Metrics

All fairness indexes mentioned above assume maximum fairness, e.g., yield a value of 1, only if the same capacity is assigned to all end-to-end demands. Dependent on the fairness method being

used, e.g., BSF where fair distribution of resources is focused on a link and not end-to-end, the fairness index may yield a value smaller than 1 even if the maximum possible fairness is achieved. For this reason, the fairness index results which are computed from the LP solutions are normalized by those from the algorithmic solution. In case the normalized index is $\gamma_{\text{norm}} = 1$, this proves that the LP determined an optimum solution.

The normalized versions of the aggregated allocation, Jain's fairness index and Vardalis's fairness index are defined in Equations (31) to (33):

$$Alloc_{\text{norm}} = \frac{Alloc_{\text{LP}}}{Alloc_{\text{Algorithm}}},\tag{31}$$

$$\gamma_{\text{Jain,norm}} = \frac{\gamma_{\text{Jain,LP}}}{\gamma_{\text{Jain,Algorithm}}},\tag{32}$$

$$\gamma_{\text{Vardalis,norm}} = \frac{\gamma_{\text{Vardalis,LP}}}{\gamma_{\text{Vardalis,Algorithm}}}.\tag{33}$$

### 4.5. Product of Efficiency and Jain's Fairness Index

It was discussed earlier that efficiency and fairness are related by a trade-off. It is useful to define a performance metric which takes both metrics into account, therefore the product of the aggregate capacity and the well-known Jain's fairness index $Alloc \cdot \gamma_{\text{Jain}}$ are considered as well in this analysis:

$$Alloc \cdot \gamma_{\text{Jain}} = \sum_{[s,t] \in K} \varphi^{st} \cdot \frac{(\sum_{[s,t] \in K} \varphi^{st})^2}{|K| \cdot \sum_{[s,t] \in K} (\varphi^{st})^2}.\tag{34}$$

## 5. Performance Evaluation

This section includes validation and performance comparisons of different allocation models. In order to validate the developed fair allocation methods, eight network topologies are considered which represent different topology types such as a linear network, a network with two alternative paths, a full-meshed network, etc. The details about the topologies are given in Figures 13–20. In the figures, the blue circles identify the nodes, the connectors represent the links and the arrow in each link specifies the direction of the data flow if the link is used. The numbers besides the connectors specify the respective link speeds.

For each of the topologies, different scenarios are created by using different nodes as senders and receivers for data flows and in some cases the capacities of individual links are changed, resulting in 31 scenarios. The link speeds are given in the respective topology figures, whereas the data flows injected into the topologies are given in Tables 2–9. The same tables also show the end-to-end throughput which each flow can achieve within a given scenario considering the different fairness methods discussed in Section 3.2. The NFF column in the tables shows the results for the NFF-objective-VI function specified in Equations (25) and (26).

The computation of the linear programming solutions is performed with the AIMMS (non-)linear programming software system [33]. The formulations for BSF and BFF are specified as a mixed-integer linear programming (MILP) problem (Source code available on [34]) which is processed by the CPLEX solver included in the AIMMS system. The NFF solution contains nonlinear components and is computed using AIMMS's Outer Approximation (AOA) module.

The AIMMS software does, however, not support batch jobs, the problem description had to be entered manually into a GUI, so that an automated evaluation of different network scenarios could not be performed. For this reason, the evaluation is performed with a selected number of scenarios.

**Table 2.** Flow set and linear programming (LP) results for the flows' end-to-end throughput for scenarios 1 to 4 described in Figure 13. All values in Mbps.

| Scenario 1 | | | | | |
| --- | --- | --- | --- | --- | --- |
| Flow | BSF I | BSF II | BFF I | BFF II | NFF |
| [a,f] | 25.0 | 25.0 | 25.0 | 25.0 | 25.0 |
| [m,n] | 10.0 | 10.0 | 10.0 | 10.0 | 10.0 |
| **Scenario 2** | | | | | |
| Flow | BSF I | BSF II | BFF I | BFF II | NFF |
| [a,f] | 18.3 | 18.3 | 10.0 | 18.3 | 12.5 |
| [a,c] | 8.3 | 8.3 | 15.0 | 8.3 | 12.5 |
| [m,n] | 8.3 | 8.3 | 10.0 | 8.3 | 10.0 |
| **Scenario 3** | | | | | |
| Flow | BSF I | BSF II | BFF I | BFF II | NFF |
| [n,h] | 7.5 | 15.8 | 7.5 | 15.8 | 10.0 |
| [m,h] | 15.0 | 8.3 | 15.0 | 8.3 | 10.0 |
| [a,h] | 7.5 | 7.5 | 7.5 | 7.5 | 10.0 |
| [a,d] | 10.0 | 8.3 | 10.0 | 8.3 | 10.0 |
| **Scenario 4** | | | | | |
| Flow | BSF I | BSF II | BFF I | BFF II | NFF |
| [a,f] | 8.3 | 13.3 | 5.0 | 13.3 | 8.8 |
| [a,c] | 8.3 | 8.3 | 15.0 | 8.3 | 8.8 |
| [a,e] | 10.0 | 5.0 | 5.0 | 5.0 | 8.8 |
| [m,n] | 8.3 | 8.3 | 10.0 | 8.3 | 8.8 |

BSF: Bottleneck Subflow Fair; BFF: Bottleneck Flow Fair; BSF: Network Flow Fair.

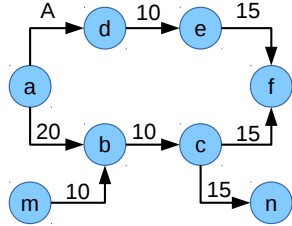

**Figure 13.** Topology of Scenarios 1 to 4: Two-path connection with shared bottleneck (b, c) on the path between **a** and **f**. The numbers specify the link speeds in Mbps. scenarios 1, 2, 4: A = 20; scenario 3: A = 15.

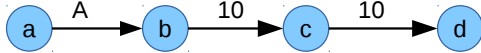

**Figure 14.** Topology of scenarios 5 to 8: all stations in a line. The numbers specify the link speeds in Mbps. Scenarios 5 and 6: A = 10; Scenarios 7 and 8: A = 20.

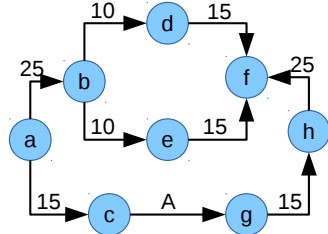

**Figure 15.** Topology of scenarios 9 to 14: Two-path connection between a and h where one of the paths is again split into two sub-paths. The numbers specify the link speeds in Mbps.

**Table 3.** Flow set and LP results for the flows' end-to-end throughput for scenarios 5 to 8 described in Figure 14. All values in Mbps.

| Scenario 5 | | | | | |
|---|---|---|---|---|---|
| Flow | BSF I | BSF II | BFF I | BFF II | NFF |
| [a,b] | 5.0 | 5.0 | 5.0 | 5.0 | 5.0 |
| [a,d] | 5.0 | 5.0 | 5.0 | 5.0 | 5.0 |
| [c,d] | 5.0 | 5.0 | 5.0 | 5.0 | 5.0 |
| **Scenario 6** | | | | | |
| Flow | BSF I | BSF II | BFF I | BFF II | NFF |
| [a,c] | 3.3 | 3.3 | 3.3 | 3.3 | 3.3 |
| [a,d] | 3.3 | 3.3 | 3.3 | 3.3 | 3.3 |
| [b,d] | 3.3 | 3.3 | 3.3 | 3.3 | 3.3 |
| **Scenario 7** | | | | | |
| Flow | BSF I | BSF II | BFF I | BFF II | NFF |
| [a,b] | 15.0 | 15.0 | 15.0 | 15.0 | 15.0 |
| [a,d] | 5.0 | 5.0 | 5.0 | 5.0 | 5.0 |
| [c,d] | 5.0 | 5.0 | 5.0 | 5.0 | 5.0 |
| **Scenario 8** | | | | | |
| Flow | BSF I | BSF II | BFF I | BFF II | NFF |
| [a,c] | 3.3 | 3.3 | 3.3 | 3.3 | 3.3 |
| [a,d] | 3.3 | 3.3 | 3.3 | 3.3 | 3.3 |
| [b,d] | 3.3 | 3.3 | 3.3 | 3.3 | 3.3 |

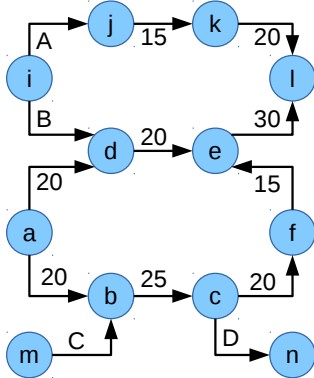

**Figure 16.** Topology of scenarios 15 to 18: 2 two-path connections i-l and a–f with shared bottleneck links d–e and b–c. The numbers specify the link speeds in Mbps. Scenarios 15 and 16: A = B = C = 10, D = 15; scenarios 17 and 18: A = 30, B = 40, C = 10, D = 35.

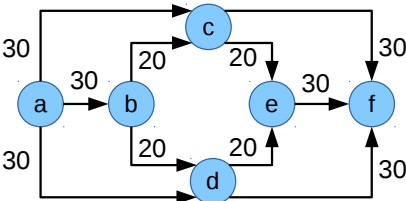

**Figure 17.** Topology of scenarios 19 and 20: Three alternative connections for flow [a, f] via c, b or d. The numbers specify the link speeds in Mbps.

**Table 4.** Flow set and LP results for the flows' end-to-end throughput for scenarios 9 to 14 described in Figure 15. All values in Mbps.

| Scenario 9 | | | | | |
| --- | --- | --- | --- | --- | --- |
| Flow | BSF I | BSF II | BFF I | BFF II | NFF |
| [a,f] | 10.0 | 15.0 | 10.0 | 15.0 | 15.0 |
| [a,f] | 20.0 | 15.0 | 20.0 | 15.0 | 15.0 |

| Scenario 10 | | | | | |
| --- | --- | --- | --- | --- | --- |
| Flow | BSF I | BSF II | BFF I | BFF II | NFF |
| [a,f] | 10.0 | 15.0 | 10.0 | 15.0 | 10.0 |
| [a,h] | 10.0 | 10.0 | 15.0 | 10.0 | 10.0 |
| [c,g] | 10.0 | 5.0 | 5.0 | 5.0 | 10.0 |

| Scenario 11 | | | | | |
| --- | --- | --- | --- | --- | --- |
| Flow | BSF I | BSF II | BFF I | BFF II | NFF |
| [a,b] | 15.0 | 8.3 | 15.0 | 10.0 | 8.8 |
| [a,f] | 10.0 | 16.7 | 5.0 | 10.0 | 8.8 |
| [a,h] | 5.0 | 5.0 | 10.0 | 10.0 | 8.8 |
| [c,g] | 5.0 | 5.0 | 5.0 | 5.0 | 8.8 |

| Scenario 12 | | | | | |
| --- | --- | --- | --- | --- | --- |
| Flow | BSF I | BSF II | BFF I | BFF II | NFF |
| [a,b] | 10.0 | 17.5 | 17.5 | 17.5 | 12.5 |
| [a,f] | 15.0 | 7.5 | 7.5 | 7.5 | 12.5 |

| Scenario 13 | | | | | |
| --- | --- | --- | --- | --- | --- |
| Flow | BSF I | BSF II | BFF I | BFF II | NFF |
| [a,f] | 20.0 | 20.0 | 20.0 | 20.0 | 20.0 |
| [c,h] | 15.0 | 15.0 | 15.0 | 15.0 | 15.0 |

| Scenario 14 | | | | | |
| --- | --- | --- | --- | --- | --- |
| Flow | BSF I | BSF II | BFF I | BFF II | NFF |
| [a,f] | 8.3 | 8.3 | 5.0 | 8.3 | 8.0 |
| [a,h] | 8.3 | 10.0 | 10.0 | 10.0 | 8.0 |
| [a,g] | 10.0 | 6.7 | 10.0 | 6.7 | 8.0 |
| [c,h] | 10.0 | 6.7 | 10.0 | 6.7 | 8.0 |
| [b,h] | 8.3 | 8.3 | 5.0 | 8.3 | 8.0 |

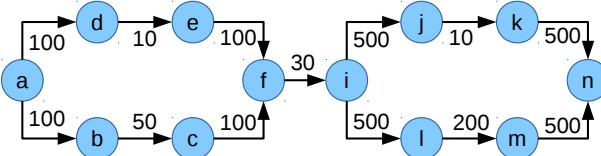

**Figure 18.** Topology of scenarios 21 to 24: Two consecutive two-path connections a-f and i-n with intermediate bottleneck link f-i. The numbers specify the link speeds in Mbps.

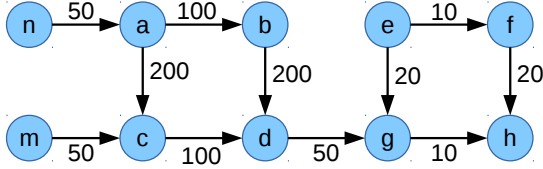

**Figure 19.** Topology of scenarios 25 to 28: Meshed network with shared bottleneck link d-g. The numbers specify the link speeds in Mbps.

**Table 5.** Flow set and LP results for the flows' end-to-end throughput for scenarios 15 to 18 described in Figure 16. All values in Mbps.

| Scenario 15 | | | | | |
|---|---|---|---|---|---|
| Flow | BSF I | BSF II | BFF I | BFF II | NFF |
| [i,l] | 20.0 | 20.0 | 20.0 | 20.0 | 20.0 |
| [a,f] | 25.0 | 25.0 | 25.0 | 25.0 | 25.0 |
| [m,n] | 10.0 | 10.0 | 10.0 | 10.0 | 10.0 |
| **Scenario 16** | | | | | |
| Flow | BSF I | BSF II | BFF I | BFF II | NFF |
| [i,l] | 15.0 | 15.0 | 10.0 | 15.0 | 10.0 |
| [i,e] | 5.0 | 5.0 | 10.0 | 5.0 | 10.0 |
| [a,c] | 15.0 | 8.3 | 15.0 | 8.3 | 12.5 |
| [a,f] | 10.0 | 8.3 | 10.0 | 8.3 | 12.5 |
| [m,n] | 10.0 | 8.3 | 10.0 | 8.3 | 10.0 |
| **Scenario 17** | | | | | |
| Flow | BSF I | BSF II | BFF I | BFF II | NFF |
| [i,l] | 25.0 | 25.0 | 25.0 | 25.0 | 20.0 |
| [a,f] | 10.0 | 22.5 | 10.0 | 22.5 | 20.0 |
| [m,n] | 25.0 | 12.5 | 25.0 | 12.5 | 20.0 |
| **Scenario 18** | | | | | |
| Flow | BSF I | BSF II | BFF I | BFF II | NFF |
| [i,l] | 15.0 | 21.7 | 15.0 | 21.7 | 15.0 |
| [i,e] | 10.0 | 6.7 | 10.0 | 6.7 | 6.3 |
| [a,c] | 12.5 | 8.3 | 12.5 | 8.3 | 11.3 |
| [a,f] | 10.0 | 15.0 | 10.0 | 15.0 | 11.3 |
| [m,n] | 12.5 | 8.3 | 12.5 | 8.3 | 11.3 |

**Table 6.** Flow set and LP results for the flows' end-to-end throughput for scenarios 19 and 20 described in Figure 17. All values in Mbps.

| Scenario 19 | | | | | |
|---|---|---|---|---|---|
| Flow | BSF I | BSF II | BFF I | BFF II | NFF |
| [a,f] | 53.3 | 60.0 | 60.0 | 50.0 | 40.0 |
| [b,f] | 33.3 | 30.0 | 30.0 | 35.0 | 40.0 |
| [a,e] | 33.3 | 30.0 | 30.0 | 35.0 | 40.0 |
| **Scenario 20** | | | | | |
| Flow | BSF I | BSF II | BFF I | BFF II | NFF |
| [a,c] | 30.0 | 23.3 | 40.0 | 20.0 | 22.5 |
| [a,d] | 30.0 | 21.7 | 10.0 | 30.0 | 22.5 |
| [a,e] | 10.0 | 28.3 | 10.0 | 20.0 | 22.5 |
| [a,f] | 20.0 | 16.7 | 30.0 | 20.0 | 22.5 |
| [b,f] | 30.0 | 30.0 | 30.0 | 30.0 | 30.0 |

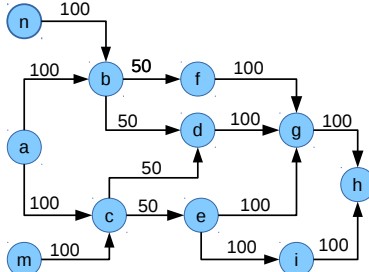

**Figure 20.** Scenario 8—Meshed network. The numbers specify the link speeds in Mbps.

**Table 7.** Flow set and LP results for the flows' end-to-end throughput for scenarios 21 to 24 described in Figure 18. All values in Mbps.

| | | | Scenario 21 | | |
|---|---|---|---|---|---|
| Flow | BSF I | BSF II | BFF I | BFF II | NFF |
| [a,f] | 60.0 | 60.0 | 60.0 | 60.0 | 60.0 |
| [i,n] | 210.0 | 210.0 | 210.0 | 210.0 | 210.0 |
| [f,i] | 30.0 | 30.0 | 30.0 | 30.0 | 30.0 |

| | | | Scenario 22 | | |
|---|---|---|---|---|---|
| Flow | BSF I | BSF II | BFF I | BFF II | NFF |
| [a,i] | 5.0 | 5.0 | 10.0 | 5.0 | 15.0 |
| [a,f] | 55.0 | 55.0 | 50.0 | 55.0 | 45.0 |
| [f,n] | 10.0 | 5.0 | 10.0 | 5.0 | 15.0 |
| [i,n] | 200.0 | 205.0 | 200.0 | 205.0 | 195.5 |

| | | | Scenario 23 | | |
|---|---|---|---|---|---|
| Flow | BSF I | BSF II | BFF I | BFF II | NFF |
| [a,i] | 20.0 | 21.7 | 5.0 | 21.7 | 15.0 |
| [a,c] | 20.0 | 16.7 | 50.0 | 16.7 | 22.5 |
| [a,f] | 20.5 | 21.7 | 5.0 | 21.7 | 22.5 |
| [f,n] | 10.0 | 3.3 | 5.0 | 3.3 | 15.0 |
| [i,n] | 190.0 | 203.3 | 200.0 | 203.3 | 185.0 |
| [i,k] | 10.0 | 3.3 | 5.0 | 3.3 | 10.0 |

| | | | Scenario 24 | | |
|---|---|---|---|---|---|
| Flow | BSF I | BSF II | BFF I | BFF II | NFF |
| [a,e] | 5.0 | 5.0 | 5.0 | 5.0 | 20.0 |
| [a,c] | 50.0 | 25.0 | 50.0 | 25.0 | 20.0 |
| [a,f] | 5.0 | 30.0 | 5.0 | 30.0 | 20.0 |
| [i,n] | 100.0 | 105.0 | 100.0 | 105.3 | 70.0 |
| [i,m] | 100.0 | 100.0 | 100.0 | 100.0 | 70.0 |
| [i,k] | 10.0 | 5.0 | 10.0 | 5.0 | 70.0 |

The multipath fair capacity allocation models developed in Section 3 are based on fair capacity allocation to individual flows utilizing maximum possible network capacity. It is essential to validate the developed models confirming the optimum solution. If the developed models provide the ideal fair allocation for any scenario, then the models can be confirmed as the optimum fair solution. As an alternative solution which complements the linear programing approach, algorithms which provide an ideal solution for a network scenario for different fair allocation methods have been proposed in [1]. These algorithms resemble the way how one would solve the problem "manually" when looking at it.

The values of the performance metrics for BSF, BFF and NFF are listed in Tables 10–14. For all investigated scenarios, the performance metrics yield the same value for both the linear programming solution and the algorithmic solution, therefore the results from the algorithmic solution are not given separately. The identity of the LP and the algorithm values confirms the linear programming model as the optimum solution. However, fairness indexes which measure the fairness performance are biased to the equal allocation to the flows and neglect the fairness criteria of the individual allocation methods. Therefore, as already mentioned in Section 4.4, the fairness indexes computed by LP need to be normalized with respect to the respective algorithmic fairness indexes. These normalized values are shown in Figures 21–24.

**Table 8.** Flow set and LP results for the flows' end-to-end throughput for scenarios 25 to 28 described in Figure 19. All values in Mbps.

| Scenario 25 | | | | | |
|---|---|---|---|---|---|
| Flow | BSF I | BSF II | BFF I | BFF II | NFF |
| [a,d] | 195.0 | 195.0 | 195.0 | 195.0 | 190.0 |
| [e,h] | 15.0 | 15.0 | 15.0 | 15.0 | 10.0 |
| [m,h] | 5.0 | 5.0 | 5.0 | 5.0 | 10.0 |
| **Scenario 26** | | | | | |
| Flow | BSF I | BSF II | BFF I | BFF II | NFF |
| [a,d] | 193.3 | 193.3 | 193.3 | 193.3 | 190.0 |
| [e,h] | 13.3 | 13.3 | 13.3 | 13.3 | 10.0 |
| [m,h] | 3.3 | 3.3 | 3.3 | 3.3 | 5.0 |
| [n,h] | 3.3 | 3.3 | 3.3 | 3.3 | 5.0 |
| **Scenario 27** | | | | | |
| Flow | BSF I | BSF II | BFF I | BFF II | NFF |
| [a,d] | 150.0 | 150.0 | 150.0 | 150.0 | 150.0 |
| [m,d] | 50.0 | 50.0 | 50.0 | 50.0 | 50.0 |
| [e,h] | 20.0 | 20.0 | 20.0 | 20.0 | 20.0 |
| **Scenario 28** | | | | | |
| Flow | BSF I | BSF II | BFF I | BFF II | NFF |
| [a,d] | 100.0 | 100.0 | 100.0 | 100.0 | 100.0 |
| [m,d] | 50.0 | 50.0 | 50.0 | 50.0 | 50.0 |
| [n,b] | 50.0 | 50.0 | 50.0 | 50.0 | 50.0 |
| [e,h] | 15.0 | 15.0 | 10.0 | 15.0 | 10.0 |
| [g,h] | 5.0 | 5.0 | 10.0 | 5.0 | 10.0 |

**Table 9.** Flow set and LP results for the flows' end-to-end throughput for scenarios 29 to 31 described in Figure 20. All values in Mbps.

| Scenario 29 | | | | | |
|---|---|---|---|---|---|
| Flow | BSF I | BSF II | BFF I | BFF II | NFF |
| [n,h] | 25.0 | 25.0 | 25.0 | 25.0 | 50.0 |
| [m,h] | 25.0 | 50.0 | 25.0 | 50.0 | 50.0 |
| [a,h] | 100.0 | 75.0 | 100.0 | 75.0 | 50.0 |
| **Scenario 30** | | | | | |
| Flow | BSF I | BSF II | BFF I | BFF II | NFF |
| [n,h] | 25.0 | 25.0 | 50.0 | 75.0 | 50.0 |
| [m,h] | 25.0 | 41.7 | 50.0 | 25.0 | 50.0 |
| [a,h] | 75.0 | 66.7 | 50.0 | 25.0 | 50.0 |
| [a,d] | 75.0 | 66.7 | 50.0 | 75.0 | 50.0 |
| **Scenario 31** | | | | | |
| Flow | BSF I | BSF II | BFF I | BFF II | NFF |
| [n,h] | 50.0 | 16.7 | 25.0 | 16.7 | 33.3 |
| [m,h] | 16.7 | 20.0 | 25.0 | 41.7 | 33.3 |
| [a,h] | 16.7 | 70.0 | 50.0 | 66.7 | 33.3 |
| [a,d] | 16.7 | 66.7 | 50.0 | 41.7 | 33.3 |
| [m,e] | 50.0 | 10.0 | 25.0 | 16.7 | 33.3 |
| [n,d] | 50.0 | 16.7 | 25.0 | 16.7 | 33.3 |

**Table 10.** Performance evaluation for BSF-I. *Alloc* values in Mbps.

| Scenario | *Alloc* | $\gamma_{\text{Jain}}$ | $\gamma_{\text{Vard}}$ | $Alloc \cdot \gamma_{\text{Jain}}$ |
|---|---|---|---|---|
| 1 | 35.0 | 0.845 | 0.571 | 29.6 |
| 2 | 35.0 | 0.860 | 0.714 | 30.1 |
| 3 | 40.0 | 0.914 | 0.833 | 36.6 |
| 4 | 35.0 | 0.993 | 0.952 | 34.7 |
| 5 | 15.0 | 1.000 | 1.000 | 15.0 |
| 6 | 10.0 | 1.000 | 1.000 | 10.0 |
| 7 | 25.0 | 0.758 | 0.600 | 19.0 |
| 8 | 10.0 | 1.000 | 1.000 | 10.0 |
| 9 | 30.0 | 0.900 | 0.667 | 27.0 |
| 10 | 30.0 | 1.000 | 1.000 | 30.0 |
| 11 | 35.0 | 0.817 | 0.714 | 28.6 |
| 12 | 25.0 | 0.962 | 0.800 | 24.1 |
| 13 | 35.0 | 0.980 | 0.857 | 34.3 |
| 14 | 40.0 | 0.914 | 0.854 | 36.6 |
| 15 | 55.0 | 0.896 | 0.773 | 49.3 |
| 16 | 55.0 | 0.896 | 0.818 | 49.3 |
| 17 | 60.0 | 0.889 | 0.750 | 53.3 |
| 18 | 60.0 | 0.976 | 0.917 | 58.6 |
| 19 | 120.0 | 0.947 | 0.833 | 113.6 |
| 20 | 120.0 | 0.900 | 0.813 | 108.0 |
| 21 | 300.0 | 0.617 | 0.450 | 185.1 |
| 22 | 270.0 | 0.422 | 0.346 | 113.9 |
| 23 | 270.0 | 0.324 | 0.356 | 87.5 |
| 24 | 270.0 | 0.536 | 0.489 | 144.7 |
| 25 | 215.0 | 0.403 | 0.140 | 86.6 |
| 26 | 213.3 | 0.303 | 0.125 | 64.6 |
| 27 | 220.0 | 0.635 | 0.477 | 139.7 |
| 28 | 220.0 | 0.635 | 0.614 | 139.7 |
| 29 | 150.0 | 0.667 | 0.500 | 100.1 |
| 30 | 200.0 | 0.800 | 0.667 | 160.0 |
| 31 | 200.0 | 0.800 | 0.700 | 160.0 |

**Table 11.** Performance evaluation for BSF-II. *Alloc* values in Mbps.

| Scenario | *Alloc* | $\gamma_{\text{Jain}}$ | $\gamma_{\text{Vard}}$ | $Alloc \cdot \gamma_{\text{Jain}}$ |
|---|---|---|---|---|
| 1 | 35 | 0.845 | 0.571 | 29.6 |
| 2 | 35 | 0.860 | 0.714 | 30.1 |
| 3 | 40 | 0.897 | 0.806 | 35.9 |
| 4 | 35 | 0.896 | 0.825 | 31.4 |
| 5 | 15 | 1.000 | 1.000 | 15.0 |
| 6 | 10 | 1.000 | 1.000 | 10.0 |
| 7 | 25 | 0.758 | 0.600 | 19.0 |
| 8 | 10 | 1.000 | 1.000 | 10.0 |
| 9 | 30 | 1.000 | 1.000 | 30.0 |
| 10 | 30 | 0.857 | 0.750 | 25.7 |
| 11 | 35 | 0.771 | 0.698 | 27.0 |
| 12 | 25 | 0.862 | 0.600 | 21.6 |
| 13 | 35 | 0.980 | 0.857 | 34.3 |
| 14 | 40 | 0.976 | 0.917 | 39.0 |
| 15 | 55 | 0.896 | 0.773 | 49.3 |
| 16 | 55 | 0.835 | 0.742 | 45.9 |
| 17 | 60 | 0.932 | 0.813 | 55.9 |
| 18 | 60 | 0.820 | 0.736 | 49.2 |
| 19 | 120 | 0.889 | 0.750 | 106.7 |
| 20 | 120 | 0.962 | 0.892 | 115.4 |

**Table 11.** *Cont.*

| Scenario | *Alloc* | $\gamma_{\text{Jain}}$ | $\gamma_{\text{Vard}}$ | *Alloc* $\cdot \gamma_{\text{Jain}}$ |
|:---:|:---:|:---:|:---:|:---:|
| 21 | 300 | 0.617 | 0.450 | 185.1 |
| 22 | 270 | 0.404 | 0.321 | 109.1 |
| 23 | 270 | 0.285 | 0.296 | 77.0 |
| 24 | 270 | 0.538 | 0.489 | 145.3 |
| 25 | 215 | 0.403 | 0.140 | 86.6 |
| 26 | 213 | 0.303 | 0.125 | 64.6 |
| 27 | 220 | 0.635 | 0.477 | 139.7 |
| 28 | 220 | 0.635 | 0.614 | 139.7 |
| 29 | 150 | 0.667 | 0.500 | 100.1 |
| 30 | 200 | 0.889 | 0.778 | 177.8 |
| 31 | 200 | 0.641 | 0.580 | 128.2 |

**Table 12.** Performance evaluation for BFF-I. *Alloc* values in Mbps.

| Scenario | *Alloc* | $\gamma_{\text{Jain}}$ | $\gamma_{\text{Vard}}$ | *Alloc* $\cdot \gamma_{\text{Jain}}$ |
|:---:|:---:|:---:|:---:|:---:|
| 1 | 35 | 0.845 | 0.571 | 29.6 |
| 2 | 35 | 0.961 | 0.857 | 33.6 |
| 3 | 40 | 0.914 | 0.833 | 36.6 |
| 4 | 35 | 0.817 | 0.714 | 28.6 |
| 5 | 15 | 1.000 | 1.000 | 15.0 |
| 6 | 10 | 1.000 | 1.000 | 10.0 |
| 7 | 25 | 0.758 | 0.600 | 19.0 |
| 8 | 10 | 1.000 | 1.000 | 10.0 |
| 9 | 30 | 0.900 | 0.667 | 27.0 |
| 10 | 30 | 0.857 | 0.750 | 25.7 |
| 11 | 35 | 0.817 | 0.714 | 28.6 |
| 12 | 25 | 0.862 | 0.600 | 21.6 |
| 13 | 35 | 0.980 | 0.857 | 34.3 |
| 14 | 40 | 0.914 | 0.813 | 36.6 |
| 15 | 55 | 0.896 | 0.773 | 49.3 |
| 16 | 55 | 0.968 | 0.909 | 53.2 |
| 17 | 60 | 0.889 | 0.750 | 53.3 |
| 18 | 60 | 0.976 | 0.917 | 58.6 |
| 19 | 120 | 0.889 | 0.750 | 106.7 |
| 20 | 120 | 0.800 | 0.708 | 96.0 |
| 21 | 300 | 0.617 | 0.450 | 185.1 |
| 22 | 270 | 0.427 | 0.346 | 115.3 |
| 23 | 270 | 0.285 | 0.289 | 77.0 |
| 24 | 270 | 0.536 | 0.489 | 144.7 |
| 25 | 215 | 0.403 | 0.140 | 86.6 |
| 26 | 213 | 0.303 | 0.125 | 64.6 |
| 27 | 220 | 0.635 | 0.477 | 139.7 |
| 28 | 220 | 0.637 | 0.614 | 140.1 |
| 29 | 150 | 0.667 | 0.500 | 100.1 |
| 30 | 200 | 1.000 | 1.000 | 200.0 |
| 31 | 200 | 0.889 | 0.800 | 177.8 |

**Table 13.** Performance evaluation for BFF-II. *Alloc* values in Mbps.

| Scenario | *Alloc* | $\gamma_{\text{Jain}}$ | $\gamma_{\text{Vard}}$ | $Alloc \cdot \gamma_{\text{Jain}}$ |
|:---:|:---:|:---:|:---:|:---:|
| 1 | 35 | 0.845 | 0.571 | 29.6 |
| 2 | 35 | 0.86 | 0.714 | 30.1 |
| 3 | 40 | 0.897 | 0.806 | 35.9 |
| 4 | 35 | 0.896 | 0.825 | 31.4 |
| 5 | 15 | 1.000 | 1.000 | 15.0 |
| 6 | 10 | 1.000 | 1.000 | 10.0 |
| 7 | 25 | 0.758 | 0.600 | 19.0 |
| 8 | 10 | 1.000 | 1.000 | 10.0 |
| 9 | 30 | 1.000 | 1.000 | 30.0 |
| 10 | 30 | 0.857 | 0.750 | 25.7 |
| 11 | 35 | 0.942 | 0.857 | 33.0 |
| 12 | 25 | 0.862 | 0.600 | 21.6 |
| 13 | 35 | 0.980 | 0.857 | 34.3 |
| 14 | 40 | 0.976 | 0.917 | 39.0 |
| 15 | 55 | 0.896 | 0.773 | 49.3 |
| 16 | 55 | 0.835 | 0.742 | 45.9 |
| 17 | 60 | 0.932 | 0.813 | 55.9 |
| 18 | 60 | 0.820 | 0.736 | 49.2 |
| 19 | 120 | 0.970 | 0.875 | 116.4 |
| 20 | 120 | 0.960 | 0.875 | 115.2 |
| 21 | 300 | 0.617 | 0.450 | 185.1 |
| 22 | 270 | 0.404 | 0.321 | 109.1 |
| 23 | 270 | 0.285 | 0.296 | 77.0 |
| 24 | 270 | 0.538 | 0.489 | 145.3 |
| 25 | 215 | 0.403 | 0.140 | 86.6 |
| 26 | 213 | 0.303 | 0.125 | 64.6 |
| 27 | 220 | 0.635 | 0.477 | 139.7 |
| 28 | 220 | 0.635 | 0.614 | 139.7 |
| 29 | 150 | 0.857 | 0.750 | 128.6 |
| 30 | 200 | 0.800 | 0.667 | 160 |
| 31 | 200 | 0.762 | 0.700 | 152.4 |

Figures 21 depicts the aggregate capacity allocation of all flows in the network. All fairness methods achieve the same network utilization for the different scenarios. Exceptions are the scenarios 23, 25 and 26 where BSF and BFF yield a value which is slightly greater than 1. This means that the BSF and BFF methods are better by a small amount in this case than NFF against which the BSF and BFF performance is normalized. The topologies deployed in these scenarios include a single link which connects two subnetworks. Bottleneck-based fairness methods appear to be better able to cope with such a situation. Figures 22–24 show the results for the fairness metrics for bottleneck subflow fair and bottleneck flow fair, again normalized by the metrics for network flow fair to provide comparability between the different scenarios. Most values range between 0.8 and 1 for BSF and BFF, which means that NFF, which is always one by definition, achieves better fairness. A direct comparison between the different BSF and BFF methods shows however no significant difference. There is a slight trend that fairness degrades in the case of scenarios 29 to 31 which are all based on the complex meshed network topology shown in Figure 20, with different flow sets. Since this is one of the most complex scenarios which was investigated, it means that realizing fairness for these more complex setups is more difficult than for simpler ones.

**Table 14.** Performance evaluation for NFF. *Alloc* values in Mbps.

| scenario | *Alloc* | $\gamma_{\text{Jain}}$ | $\gamma_{\text{Vard}}$ | $Alloc \cdot \gamma_{\text{Jain}}$ |
|---|---|---|---|---|
| 1 | 35 | 0.845 | 0.571 | 29.6 |
| 2 | 35 | 0.990 | 0.929 | 34.7 |
| 3 | 40 | 1.000 | 1.000 | 40.0 |
| 4 | 35 | 1.000 | 1.000 | 35.0 |
| 5 | 15 | 1.000 | 1.000 | 15.0 |
| 6 | 10 | 1.000 | 1.000 | 10.0 |
| 7 | 25 | 0.758 | 0.600 | 18.95 |
| 8 | 10 | 1.000 | 1.000 | 10.0 |
| 9 | 30 | 1.000 | 1.000 | 30.0 |
| 10 | 30 | 1.000 | 1.000 | 30.0 |
| 11 | 35 | 1.000 | 1.000 | 35.0 |
| 12 | 25 | 1.000 | 1.000 | 25.0 |
| 13 | 35 | 0.980 | 0.857 | 34.3 |
| 14 | 40 | 1.000 | 1.000 | 40.0 |
| 15 | 55 | 0.896 | 0.773 | 49.3 |
| 16 | 55 | 0.988 | 0.932 | 54.3 |
| 17 | 60 | 1.000 | 1.000 | 60.0 |
| 18 | 60 | 0.985 | 0.938 | 59.1 |
| 19 | 120 | 1.000 | 1.000 | 120.0 |
| 20 | 120 | 0.985 | 0.938 | 118.2 |
| 21 | 300 | 0.617 | 0.450 | 185.1 |
| 22 | 270 | 0.450 | 0.370 | 121.5 |
| 23 | 260 | 0.379 | 0.361 | 98.5 |
| 24 | 270 | 0.764 | 0.667 | 206.3 |
| 25 | 210 | 0.405 | 0.143 | 85.1 |
| 26 | 210 | 0.304 | 0.127 | 63.8 |
| 27 | 220 | 0.635 | 0.477 | 139.7 |
| 28 | 220 | 0.637 | 0.614 | 140.1 |
| 29 | 150 | 1.000 | 1.000 | 150.0 |
| 30 | 200 | 1.000 | 1.000 | 200.0 |
| 31 | 200 | 1.000 | 1.000 | 200.0 |

The examples show how optimum resource allocation and fairness can be achieved in case of optimum conditions, i.e., perfect global knowledge of the network topology and of all flow demands which are injected at different nodes. Practical congestion control methods for multipath transport are restricted to observations at the end nodes, e.g., an increase of the delay or packet losses. With this limited knowledge, they have to draw conclusions about the conditions in the network which obviously cannot be optimum. The theoretical methods can however serve as a benchmark to show how well a practical multipath transport implementation can fulfill performance and fairness requirements, in other words, how closely it approaches the optimum.

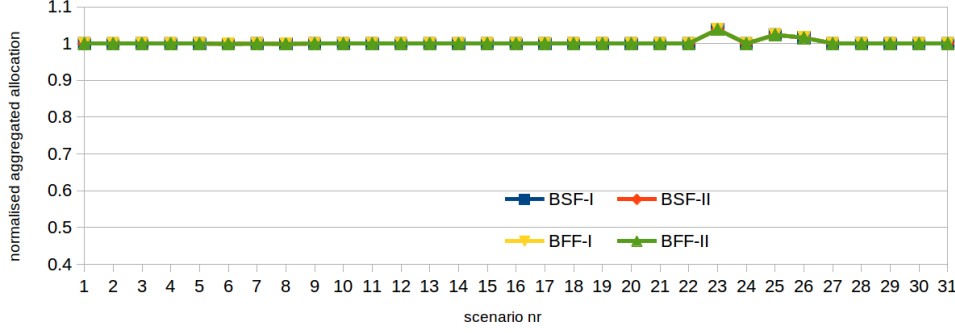

**Figure 21.** Aggregate allocation, normalized by NFF, for different fair allocation methods, based on the numerical results in Tables 10–14.

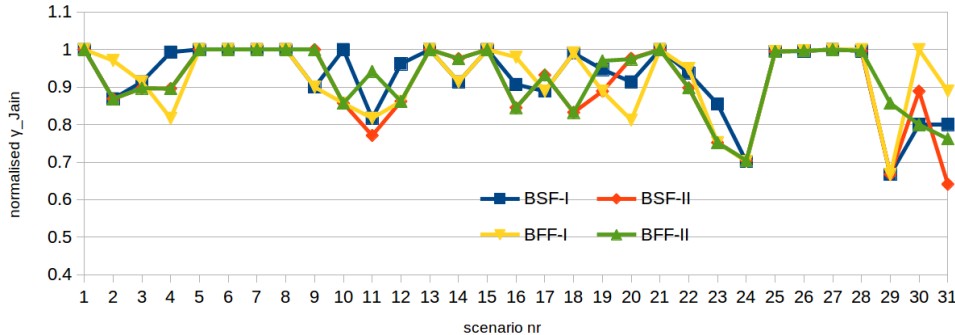

**Figure 22.** Jain's fairness index, normalized by NFF, for different fair allocation methods, based on the numerical results in Tables 10–14.

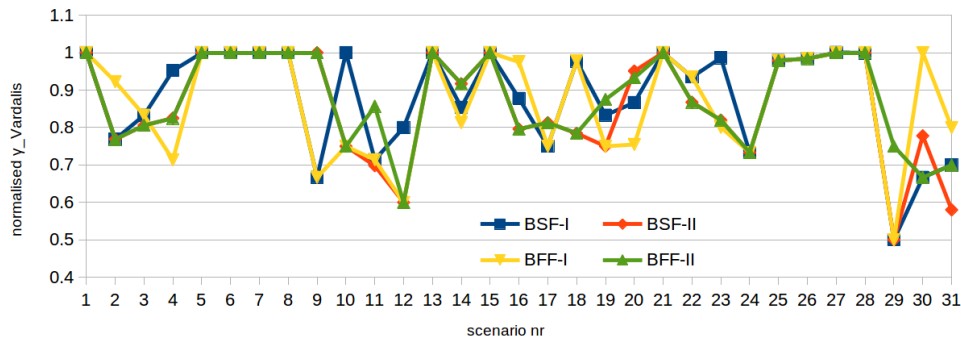

**Figure 23.** Vardalis's fairness index, normalized by NFF, for different fair allocation methods, based on the numerical results in Tables 10–14.

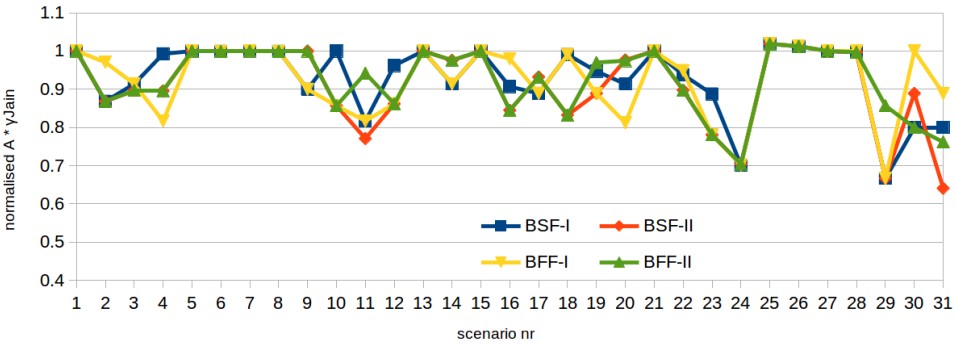

**Figure 24.** Product of efficiency and Jain's fairness index, normalized by NFF, for different fair allocation methods, based on the numerical results in Tables 10–14.

## 6. Conclusions

The aim of this work was a rigorous analysis of fairness in multipath transport. Formal definitions for the different resources and participants in a network were given, based on which fairness definitions were specified. An optimum solution for the resource allocation in arbitrary networks considering both high network utilization as well as fairness was developed using mixed (non-)linear programming. The results were compared with an algorithmic solution which imitates the way resources would be assigned manually. It can be observed that the network utilization, measured by the aggregated allocation of all flows, is almost the same for all fairness methods, whereas network-fair allocation, according to the investigated fairness metrics, performs best if fairness is in focus. The achieved results

can serve as a benchmark to assess the performance and fairness of existing MPTCP implementations. Since the best fairness results can be achieved with the network-based solution and an individual (MP)TCP flow, this suggests that assistance of the network operator may be useful. MPTCP is an end-to-end solution; the protocol has only information about the end-to-end connection, but not about the entire network.

The presented results show the optimum assignments of data flows in the case of perfect "god-view" network knowledge and an idealized view with constant data flows. They provide a benchmark for practical implementations of multipath transport whose congestion control only has a "local" view onto the network by analysing the performance of ongoing connections.

In order to enhance the performance of the purely transport-layer based approach, it can be complemented by the network infrastructure in two ways: multipath routing which is aware of the internal network state can further optimize network utilization and fairness. Another possible approach could be that routers are equipped with intelligence to identify MPTCP data transfers. Using this data, packets could be selectively dropped in order to trigger the sender's congestion control with the target of shaping the traffic injected into the network.

The results which are given for fairness should by interpreted with care for two reasons: on the one hand, as already mentioned in the Introduction, an idealistic view onto a network with perfect global knowledge is assumed. In practice, an endpoint running a practical transport protocol can only roughly estimate possible constraints which may apply to the data flow which it injects into the network. On the other hand, the fairness metrics available in the literature which are used in this paper are designed for single path transport. The investigations in this paper show that they cannot give a direct comparison independent from the observed scenario and fairness method, which is the reason why the normalization has to be brought in. This situation is not satisfying; it would be useful if dedicated fairness metrics for multipath transport were available, which is not the case to the best of the authors' knowledge. An important step therefore would be the extension of the fairness metrics for multipath transport, which allows direct comparison of multipath fairness for any scenario irrespective of the fairness method.

**Author Contributions:** Conceptualization, A.K., M.S., A.S. and A.F.; Data Curation, M.S.; Formal Analysis, M.S. and A.S.; Funding Acquisition, A.F.; Investigation, A.K., M.S. and A.S.; Methodology, A.K., M.S., A.S. and A.F.; Project Administration, A.F.; Software, M.S.; Supervision, A.F.; Visualization, A.K.; Writing—Original Draft, A.K. and M.S.; Writing—Review and Editing, A.K.

**Funding:** This research was funded in part by Deutsche Forschungsgemeinschaft (DFG) under the grant GO 730/9-1.

**Conflicts of Interest:** The authors declare no conflict of interest.

## Appendix A. Mathematical Annex

In this annex, the mathematical description is given for the constraints of the MINLP problem which were specified in a textual way in Section 3. The symbols used in the descriptions are summarized in Table A1.

*Appendix A.1. General Constraints*

**Flow conservation constraint:** At each node except for the communication endpoints, the number of ingoing and outgoing flows must be equal [16]; routers do not act as source or sink.

$$\sum_{(i,j)\in A} f_{ij}^{st} - \sum_{(j,i)\in A} f_{ji}^{st} = \begin{cases} \varphi^{st}, & \text{if } i = s, \\ -\varphi^{st}, & \text{if } i = t, \\ 0, & \text{else,} \end{cases}$$

$$\forall i \in V, [s,t] \in K. \quad \text{(A1)}$$

**Table A1.** Symbols used in the Mixed Integer Non-Linear Programming (MINLP) model.

| | |
|---|---|
| $V$ | set of all nodes |
| $L$ | set of all interfaces |
| $A$ | set of all links |
| $K$ | set of all flows (single or multipath) |
| $(i, j)$ | link $\in A$ from interface $i$ to $j$ |
| $[s, t]$ | flow (legacy or multipath) $\in K$ from node $s$ to $t$ |
| $Alloc$ | total allocations for all flows |
| $f_i$ | legacy or MPTCP flow $i$ |
| $sf_i$ | legacy flow or MPTCP subflow $i$ |
| $f\_alloc_i$ | allocation for legacy or MPTCP flow $i$ |
| $sf\_alloc_i$ | allocation for legacy flow or MPTCP subflow $i$ |
| $flow\_diff^{st}$ | sum of the differences of the allocation to flow $[s, t]$ and any other flow |
| $\delta$ | sum of $flow\_diff_{st}$ for all pairs $s, t$ |
| $M$ | aggregated allocation for all multipath flows |
| $M_{\text{gain}}$ | Allocation gain for multipath-capable flows when upgraded from single-path to multipath |
| $cap_{ij}$ | capacity on link $(i, j)$ |
| $\varphi^{st}$ | capacity allocated to flow $[s, t]$ (single or multipath) |
| $m^{st}$ | capacity allocated to multipath flow $[s, t]$ 0 for single-path flows |
| $f_{ij}^{st}$ | capacity allocated to flow $[s, t]$ on link $(i, j)$ |
| $unused\_cap_{ij}$ | unused capacity on link $(i, j)$ |
| $max\_sf\_val^{st}$ | maximum allocation to a subflow of flow $[s, t]$ |
| $sf\_id_{ohij}^{st}$ | binary, 1 if subflow $(o, h)$ belonging to flow $[s, t]$ is present on link $(i, j)$ |
| $flow\_type^{st}$ | binary, 1 if flow $[s, t]$ is a multipath flow |
| $\beta$ | weighting factor in the objective function to prefer maximum allocation or fairness |
| $snd\_int^{st}$, $rcv\_int^{st}$ | number of sender. resp. receiver interfaces for flow $[s, t]$ |
| $flow\_id\_bin_i^{st}$ | binary, 1 if node $i$ is part of the path from node $s$ to $t$ |
| $x_{ij}^{st}$ | binary, 1 if link $(i, j)$ is in the path of flow $[s, t]$ |
| $y_{ij}^{st}$ | binary, 1 if link $(i, j)$ is a bottleneck for flow $[s, t]$ |
| $no\_of\_path_i^{st}$ | number of paths for flow $[s, t]$ passing node $i$ |
| $z_{oh}^{st}$ | binary, 1 if flow $[s, t]$ has a subflow on link $(o, h)$ |
| $splitting\_node^{st}$ | number of paths used by flow $[s, t]$ |
| $n\_sf^{st}$ | number of subflows used by flow $[s, t]$ |
| $sf\_bneck_{ohij}^{st}$ | binary, 1 if subflow $[o, h]$ of flow $[s, t]$ is congested on link $(i, j)$ |
| $cong\_flow\_diff_{ij}^{st}$ | sum of allocation differences of flow $[s, t]$ and any other flow on link $(i, j)$ |
| $cong\_group\_id_{ij}^{st}$ | binary, 1 if flow $[s, t]$ shares link $(i, j)$ with other flows |
| $SF$ | total number of subflows allocated to all flows |

MPTCP: Multipath TCP.

**Capacity limitation constraints**: The overall throughput on link $(i, j)$, summed up for all flows $[s, t]$, cannot be higher than the capacity $cap_{ij}$ of the link [16]:

$$\sum_{[s,t] \in K} f_{ij}^{st} \leq cap_{ij} \qquad \forall (i, j) \in A. \tag{A2}$$

Equation (A3) expresses the condition that a flow $[s, t] \in K$ can only occupy capacity on a link $(i, j) \in A$ if the latter is part of the flow's path. If the flow makes use of the link, the capacity allocated for the flow can at maximum be the physical speed of the link:

$$f_{ij}^{st} \leq cap_{ij} x_{ij}^{st} \qquad \forall (i, j) \in A, [s, t] \in K. \tag{A3}$$

**Path constraints:** Equation (A4) limits the number of available paths between the sender and the receiver to the product of the interfaces which the sender and the receiver are respectively equipped with. Obviously, the same limit also applies to the number of subflows which belong to the same connection.

Equation (A5) is similar to Equation (A3); however, the link and the path variables are exchanged between the left and right side of the equation sign. This means that the (non-)occupation of a link by a flow is both a necessary as well as a sufficient criterion for the fact that the link is part of the flow:

$$\sum_{(i,j) \in A} x_{ij}^{st} \leq snd\_int^{st} \cdot rcv\_int^{st} \qquad \forall i \in V, [s, t] \in K, \tag{A4}$$

$$x_{ij}^{st} \leq \beta \cdot f_{ij}^{st} \qquad \forall i, j \in A, [s, t] \in K. \tag{A5}$$

**Multipath flow identifier constraints:** Equations (A6) and (A7) are complementary: the further forces the binary variable $flow\_id\_bin$ to 0 if node $i$ is *not* part of a path, which is assigned to the flow from node $s$ to $t$; the latter forces the variable to be greater than zero if node $i$ is part of such a path.

Equation (A8) makes sure that the variable $flow\_type^{st} = 0$ if the flow $[s, t] \in K$ does not take multipath from any node $i$. Equation (A9) identifies the multipath flow i.e., $flow\_type^{st} = 1$ if the flow $[s, t] \in K$ takes multipath from any node $i$:

$$flow\_id\_bin_i^{st} \leq no\_of\_path_i^{st} \qquad \forall i \in V, [s, t] \in K, \tag{A6}$$

$$\beta \cdot flow\_id\_bin_i^{st} \geq no\_of\_path_i^{st} \qquad \forall i \in V, [s, t] \in K, \tag{A7}$$

$$flow\_type^{st} \leq \sum_{i \in V} (no\_of\_path_i^{st} - flow\_id\_bin_i^{st}) \qquad \forall [s, t] \in K, \tag{A8}$$

$$\beta \cdot flow\_type^{st} \geq \sum_{i \in V} (no\_of\_path_i^{st} - flow\_id\_bin_i^{st}) \qquad \forall [s, t] \in K. \tag{A9}$$

**Multipath subflow identification constraints:** Let $sf\_id_{ohij}^{st}$ be a binary variable that identifies subflow $[o, h] \in A$ on link $(i, j) \in A$ for the flow $[s, t] \in K$. The binary variable $sf\_id_{ohij}^{st} = 1$ if there exists a subflow $(o, h) \in A$ on link $(i, j) \in A$ for the flow $[s, t] \in K$. There is no subflow for the single path flow $[s, t] \in K$ so $sf\_id_{ohij}^{st} = 0$. Equation (A10) implies that there can not be a subflow on link $(i, j) \in A$ for the flow $[s, t] \in K$ i.e., $sf\_id_{ohij}^{st} = 0$ if the flow $[s, t] \in K$ does not use ($x_{ij}^{st} = 0$) that link. Equation (A11) generates at least one subflow for the multipath flow ($flow\_type^{st} = 1$) $[s, t] \in K$ on the link $(i, j) \in A$ if the multipath flow $[s, t] \in K$ uses that path. Equation (A12) makes sure that, for the subflow $[o, h] \in A$, the subflow identifier $sf\_id_{ohij}^{st}$ is not set to more than one link from a certain node $i$ for the flow $[s, t] \in K$:

$$sf\_id_{ohij}^{st} \leq x_{ij}^{st} \quad \forall h \neq s, (o, h) \in A, (i, j) \in A, [s, t] \in K, \tag{A10}$$

$$\sum_{(o,h) \in A} sf\_id^{st}_{ohij} \geq x^{st}_{ij} + 1 - flow\_type^{st} \qquad \forall h \neq s, (i,j) \in A, [s,t] \in K, \tag{A11}$$

$$\sum_{(i,j) \in A} sf\_id^{st}_{ohij} \leq 1 \qquad \forall i \in V, i \neq t, h \neq s, (o,h) \in A, [s,t] \in K. \tag{A12}$$

Consider a helping binary variable $z^{st}_{oh}$ which is 1 if there exists a subflow $[o,h] \in A$ on link $(o,h) \in A$ for the flow $[s,t] \in K$. $z^{st}_{oh}$ identifies the subflow $[o,h] \in A$ for the flow $[s,t] \in K$ not considering the routing path for the subflow. Thus, the information through which links the subflow is routed can not be identified from $z^{st}_{oh}$ which can be found through subflow identifier $sf\_id^{st}_{ohij}$:

$$z^{st}_{oh} = sf\_id^{st}_{ohoh} \qquad \forall h \neq s, (o,h) \in A, [s,t] \in K. \tag{A13}$$

Equation (A14) sets the variable $sf\_id^{st}_{ihih}$ to zero if the flow starting at node $i$ does not split into subflows, i.e., the flow only uses one single path to the destination. Equation (A15) signifies that, if there is no subflow $[o,h] \in A$ for the flow $[s,t] \in K$, i.e., the $z^{st}_{oh} = 0$, the subflow identifier $sf\_id^{st}_{ohij}$ should be set to zero for any link $(i,j) \in A$. Once a subflow is initialized from a certain node i.e., $sf\_id^{st}_{ohij}$ is set to 1, Equation (A16) extends the subflow identification over the path it takes to reach the destination. Equation (A17) ensures that, if there is an incoming subflow $[o,h] \in A$ on link $(i,j) \in A$ to any node $j$ which is not a source or destination node, the subflow $[o,h] \in A$ also exists on the link $(j,m) \in A$ to complete the subflow path:

$$sf\_id^{st}_{ihih} \leq no\_of\_path^{st}_i - flow\_id\_bin^{st}_i \qquad \forall i \in V, h \neq s, (i,h) \in A, [s,t] \in K, \tag{A14}$$

$$z^{st}_{oh} \geq sf\_id^{st}_{ohij} \quad \forall h \neq s, (o,h) \in A, (i,j) \in A, [s,t] \in K, \tag{A15}$$

$$\sum_{(i,j) \in A} sf\_id^{st}_{ohij} = \sum_{(j,m) \in A} sf\_id^{st}_{ohjm} \qquad \forall j \in V, j \neq s, j \neq t, h \neq s, (o,h) \in A, [s,t] \in K. \tag{A16}$$

The previously introduced equations ensure that subflows are not created erroneously but do not limit the number of subflows which a single flow can be split into. In order to enforce this limit, a variable $splitting\_node^{st}$ which counts the total number of paths used by the flow $[s,t] \in K$ is defined:

$$splitting\_node^{st} = 1 + \sum_{i \in V} (no\_of\_path^{st}_i - flow\_id\_bin^{st}_i) \qquad \forall [s,t] \in K. \tag{A17}$$

Equation (A18) limits the number of paths resp. subflows for the flow $[s,t] \in K$ to the product of the sender and receiver interfaces. For each interface pair, maximum one subflow is possible:

$$splitting\_node^{st} \leq snd\_int^{st} \cdot rcv\_int^{st} \qquad \forall [s,t] \in K. \tag{A18}$$

A helping variable $n\_sf^{st}$ is introduced to express that the number of subflows is identical to the number of paths used for a multipath flow. $n\_sf^{st}$ counts the total number of subflows for the flow $[s,t] \in K$. Equations (A19)–(A21) result in $splitting\_node^{st} = \sum_{(i,j) \in A} sf\_id^{st}_{ijij} = n\_sf^{st}$ when $flow\_type^{st} = 1$:

$$splitting\_node^{st} \leq \sum_{(i,j) \in A} sf\_id^{st}_{ijij} + 1 - flow\_type^{st} \qquad \forall [s,t] \in K, \tag{A19}$$

$$1 - flow\_type^{st} + n\_sf^{st} \leq splitting\_node^{st} \qquad \forall [s,t] \in K, \tag{A20}$$

$$\sum_{(i,j) \in A} sf\_id^{st}_{ijij} \leq n\_sf^{st} \qquad \forall j \neq s, [s,t] \in K. \tag{A21}$$

Equations (A22) to (A24) prevent subflow creation if the source as well as the destination have only one interface. The equations also ensure that only one subflow can be created for any source–destination

interface pair. Equation (A22) means that, for the flow $[s,t] \in K$, both subflows $[m,n] \in A$ and $[m,p] \in A$ can not be created from the node $m$ using the same source interface $(s,j) \in A$ and destination interface $(i,t) \in A$ pair. Equation (A23) implies that there can not be two incoming subflows $(m,p) \in A$ and $(o,p) \in A$ on the link $p$ for the flow $[s,t] \in K$ using the same source interface $(s,j) \in A$ and destination interface $(i,t) \in A$ pair. Equation (A24) refers both subflows $[m,p] \in A$ and $[o,n] \in A$ can not be created from two different nodes using the same source interface $(s,j) \in A$ and destination interface $(i,t) \in A$ pair for the flow $[s,t] \in K$:

$$sf\_id^{st}_{mnsj} + sf\_id^{st}_{mpsj} + sf\_id^{st}_{mnit} + sf\_id^{st}_{mpit} \leq 3$$
$$\forall i \in V, j \in V, (i,t) \in A, (s,j) \in A, (m,n) \in A, (m,p) \in A, n \neq p, [s,t] \in K, \quad \text{(A22)}$$

$$sf\_id^{st}_{opsj} + sf\_id^{st}_{mpsj} + sf\_id^{st}_{opit} + sf\_id^{st}_{mpit} \leq 3$$
$$\forall i \in V, j \in V, (i,t) \in A, (s,j) \in A, (o,p) \in A, (m,p) \in A, o \neq m, [s,t] \in K, \quad \text{(A23)}$$

$$sf\_id^{st}_{onsj} + sf\_id^{st}_{mpsj} + sf\_id^{st}_{onit} + sf\_id^{st}_{mpit} \leq 3$$
$$\forall i \in V, j \in V, (i,t) \in A, (s,j) \in A, (o,n) \in A, (m,p) \in A, o \neq m, n \neq p, [s,t] \in K. \quad \text{(A24)}$$

**Congested links identifier constraints for flows:** Equation (A25) implies that there exists at least one congested link for each flow. Equation (A26) makes sure that the link identified to be congested is fully utilized i. e, the flow allocations uses the full capacity. If the link $(i,j) \in A$ is congested ($y^{st}_{ij} = 1$) for the demand $[s,t] \in K$, the sum of allocated capacity to the flows sharing the link $(i,j) \in A$ is equal to the capacity of the link $(i,j) \in A$. It has already been concluded that the aggregated flow allocation on a link cannot be larger than the capacity of that link. Equation (A27) relates the two binary variables $x^{st}_{ij}$ and $y^{st}_{ij}$. It expresses that the link $(i,j) \in A$ can be on the path for the flow $[s,t] \in K$ even though the link $(i,j) \in A$ is not fully utilized. It also ensures that the identified bottleneck link $(i,j) \in A$ for the flow $[s,t] \in K$ is on the allocated path of the flow $[s,t] \in K$ i.e.,, $y^{st}_{ij}$ can be 1 only when $x^{st}_{ij} = 1$:

$$\sum_{(i,j) \in A} y^{st}_{ij} \geq 1 \qquad \forall [s,t] \in K, \qquad \text{(A25)}$$

$$\sum_{[o,d] \in K} f^{od}_{ij} \geq cap_{ij} y^{st}_{ij} \qquad \forall (i,j) \in A, [s,t] \in K, \qquad \text{(A26)}$$

$$y^{st}_{ij} \leq x^{st}_{ij} \qquad \forall (i,j) \in A, [s,t] \in K. \qquad \text{(A27)}$$

**Congested links identifier constraints for multipath subflows:** If a flow splits into multiple subflows, each of them should be a part of at least one congested link, as it is the case for single-path flows. It can, however, happen that, on a given link, two or more subflows belong to the same flow. To identify the congested link for a subflow, consider a binary variable $sf\_bneck^{st}_{ohij}$ which is 1 if the subflow $[o,h] \in A$ of the flow $[s,t] \in K$ is congested on link $(i,j) \in A$. Equation (A28) refers that the link $(i,j) \in A$ can be identified as congested link for the subflow $[o,h] \in A$ of the flow $[s,t] \in K$ only if the subflow $[o,h] \in A$ exists on that link. Equation (A29) makes sure that at least one congested link $(i,j) \in A$ for the subflow $[o,h] \in A$ of the flow $[s,t] \in K$. Equation (A30) implies that the subflow $[o,h] \in A$ of the flow $[s,t] \in K$ can not be congested on $(i,j) \in A$ if the flow $[s,t] \in K$ itself is not congested on that link i.e., $sf\_bneck^{st}_{ohij} = 0$ when $y^{st}_{ij} = 0$. Equation (A31) specifies that, if flow $[s,t]$ is bottlenecked on link $(i,j)$, then all subflows $[o,h]$ of flow $[s,t]$ are bottlenecked on $(i,j)$:

$$sf\_bneck^{st}_{ohij} \leq sf\_id^{st}_{ohij} \qquad \forall h \neq s, (o,h) \in A, (i,j) \in A, [s,t] \in K, \qquad \text{(A28)}$$

$$\sum_{(i,j)\in A} sf\_bneck^{st}_{ohij} \geq z^{st}_{oh} \qquad \forall h \neq s, (o,h) \in A, [s,t] \in K, \tag{A29}$$

$$sf\_bneck^{st}_{ohij} \leq y^{st}_{ij} \qquad \forall h \neq s, (o,h) \in A, (i,j) \in A, [s,t] \in K, \tag{A30}$$

$$\beta \cdot (1 - flow\_type^{st}) + sf\_bneck^{st}_{ohij} \geq y^{st}_{ij} - \beta \cdot (1 - sf\_id^{st}_{ohij})$$
$$\forall h \neq s, (o,h) \in A, (i,j) \in A, [s,t] \in K. \tag{A31}$$

After giving the general constraints of the linear programming model which are common to all fairness methods, the constraints specific for the particular fairness methods BSF, BFF and NFF are now discussed.

*Appendix A.2. Allocation-Specific Constraints*

Appendix A.2.1. Bottleneck Subflow Fair Allocation (BSF)

**Constraints for identifying the allocation of multipath flows:** The objective function BSF-II (Equation (9)) uses the aggregated capacity allocation to all multipath flows $M$ as defined in Equation (8). As already mentioned in the description for this equation, $m^{st}$ specifies the capacity allocation to a multipath flow $[s,t]$ and is 0 for single-path flows. In contrast to this, the variable $\varphi^{st}$ is the allocation to *any* flow $[s,t] \in K$ irrespective whether the flow is a multipath flow or a single path flow. Equation (A32) sets the value of $m^{st} = 0$ for the single path flow $[s,t] \in K$. Equation (A33) and Equation (A34) make sure that, for the multipath flow $[s,t] \in K$, $m^{st}$ is the end-to-end capacity allocation for that flow i.e., $m^{st} = \varphi^{st}$ when $flow\_type^{st} = 1$:

$$m^{st} \leq \beta \cdot flow\_type^{st} \qquad \forall [s,t] \in K, \tag{A32}$$

$$m^{st} \leq \varphi^{st} \qquad \forall [s,t] \in K, \tag{A33}$$

$$m^{st} \geq \varphi^{st} - \beta \cdot (1 - flow\_type^{st}) \qquad \forall [s,t] \in K. \tag{A34}$$

**Multipath subflow allocation mapping constraints:** Let $sf\_alloc^{st}_{ohij}$ be the subflow allocation of the subflow $[o,h] \in A$ on the link $(i,j) \in A$ for the flow $[s,t] \in K$. Equation (A35) implies that the subflow $[o,h] \in A$ does not get any allocation on the link $(i,j) \in A$ if there is no subflow $[o,h] \in A$ exist on the link $(i,j) \in A$ for the flow $[s,t] \in K$ i.e., $sf\_alloc^{st}_{ohij} = 0$ when $sf\_id^{st}_{ohij} = 0$. Equation (A36) ensures that the subflow $[o,h] \in A$ gets some allocation on the link $(i,j) \in A$ when there is a subflow $[o,h] \in A$ on the link $(i,j) \in A$ for the flow $[s,t] \in K$:

$$sf\_alloc^{st}_{ohij} \leq \beta \cdot sf\_id^{st}_{ohij} \qquad \forall [o,h] \in A, (i,j) \in A, [s,t] \in K, \tag{A35}$$

$$\beta \cdot sf\_alloc^{st}_{ohij} \geq sf\_id^{st}_{ohij} \qquad \forall [o,h] \in A, (i,j) \in A, [s,t] \in K. \tag{A36}$$

Equation (A37) and Equation (A38) map the flow allocation with the sum of the subflow allocation for a flow on a specific link. For a multipath flow, on a specific link, the sum of the allocation of all subflows of a flow should be equal to the flow allocation for that flow on that link. This implies that $f^{st}_{ij} = \sum_{(o,h)\in A} sf\_alloc^{st}_{ohij}$ when $flow\_type^{st} = 1$:

$$f^{st}_{ij} \geq \sum_{(o,h)\in A} sf\_alloc^{st}_{ohij} \qquad \forall h \neq s, (i,j) \in A, [s,t] \in K, \tag{A37}$$

$$cap_{ij}(1 - flow\_type^{st}) + \sum_{(o,h)\in A} sf\_alloc^{st}_{ohij} \geq f^{s}_{ij}t \qquad \forall h \neq s, (i,j) \in A, [s,t] \in K. \tag{A38}$$

Once a subflow gets an allocation on a link, Equation (A39) makes sure that the allocation of that subflow is the same on all the other links on its path. From any link $j$ when $j$ is not a source or destination node, the allocation of the subflow $[o, h] \in A$ of the flow $[s, t] \in K$ is equal on the incoming node $(i, j) \in A$ and the outgoing node $(j, m) \in A$:

$$\sum_{(i,j) \in A} sf\_alloc_{ohij}^{st} = \sum_{(j,m) \in A} sf\_alloc_{ohjm}^{st} \qquad \forall j \in V, j \neq s, j \neq t, (o, h) \in A, (i, j) \in A, [s, t] \in K. \quad \text{(A39)}$$

**Bottleneck subflow fair allocation constraints**: Let $sf\_link\_share_{ij}$ is the maximum share of the capacity that a subflow can get from the link $(i, j) \in A$. Equation (A40) and Equation (A41) ensure equal allocation to all the subflows that are bottlenecked on the link $(i, j) \in A$. The subflow $[o, h] \in A$ gets the maximum share of the capacity on the link $(i, j) \in A$ if the link is identified as bottleneck link ($sf\_bneck_{ohij}^{st} = 1$) for that subflow. Subflow allocation $sf\_alloc_{ohij}^{st} = sf\_link\_share_{ij}$ when $sf\_bneck_{ohij}^{st} = 1$:

$$sf\_alloc_{ohij}^{st} \leq sf\_link\_share_{ij} \qquad \forall(o, h) \in A, (i, j) \in A, [s, t] \in K, \quad \text{(A40)}$$

$$sf\_alloc_{ohij}^{st} \geq sf\_link\_share_{ij} - cap_{ij}(1 - sf\_bneck_{ohij}^{st}) \qquad \forall(o, h) \in A, (i, j) \in A, [s, t] \in K. \quad \text{(A41)}$$

In bottleneck subflow fairness, single-path flows and individual subflows of mulitpath flows are equivalent and should in the ideal case get the same share of the link capacity. Equation (A42) and Equation (A43) ensure the fair share for a single path flow on the bottleneck link. A single path flow $[s, t] \in K$ gets the maximum share of the capacity on its bottleneck link $(i, j) \in A$ i.e., $f_{ij}^{st} = sf\_link\_share_{ij}$ when $flow\_type^{st} = 0$ and $y_{ij}^{st} = 1$. For a multipath flow, Equation (A42) and Equation (A43) do not have any effect on the allocation:

$$f_{ij}^{st} \leq sf\_link\_share_{ij} + cap_{ij} \cdot flow\_type^{st} \qquad \forall(i, j) \in A, [s, t] \in K, \quad \text{(A42)}$$

$$f_{ij}^{st} \geq sf\_link\_share_{ij} - cap_{ij}(1 - y_{ij}^{st}) - cap_{ij} \cdot flow\_type^{st} \qquad \forall(o, h) \in A, (i, j) \in A, [s, t] \in K. \quad \text{(A43)}$$

Appendix A.2.2. Bottleneck Flow Fair Allocation (BFF)

**Bottleneck flow fair allocation constraints:** Let $flow\_link\_share_{ij}$ is the maximum amount of the capacity a flow can be allocated on the link $(i, j) \in A$. Equations (A44) and (A45) ensure equal share to all flows which are bottlenecked on the link $(i, j) \in A$. Equation (A44) expresses $flow\_link\_share_{ij}$ as the maximum share of the capacity to a flow on the link $(i, j) \in A$. If the link $(i, j) \in A$ is bottlenecked for the flow $[s, t] \in K$ i.e., $y_{ij}^{st} = 1$, Equation (A45) ensures maximum share of the capacity $flow\_link\_share_{ij}$ to the flow $[s, t] \in K$ on that bottleneck link. If $y_{ij}^{st} = 1$, then $flow\_link\_share_{ij} = f_{ij}^{st}$ which means equal share of the bottleneck capacity to all the flows bottlenecked on the link $(i, j) \in A$:

$$f_{ij}^{st} \leq flow\_link\_share_{ij} \qquad \forall(i, j) \in A, [s, t] \in K, \quad \text{(A44)}$$

$$f_{ij}^{st} \geq flow\_link\_share_{ij} - cap_{ij}(1 - y_{ij}^{st}) \qquad \forall(o, h) \in A, (i, j) \in A, [s, t] \in K. \quad \text{(A45)}$$

Besides assigning equal share to the flows on a bottlenecked link, BFF optionally takes a second step if there are flows which are represented by more than one subflow on the link. For each of these flows, BFF tries to share the capacity assigned to the respective flow equally among its subflows. Equation (A46) and Equation (A47) make sure that all the subflows of a flow get equal share from the flow capacity on that bottleneck link. Let $sf\_link\_share_{ij}^{st}$ be the maximum allocation that a subflow of the flow $[s, t] \in K$ can get on the link $(o, h) \in A$. The difference between $sf\_link\_share_{ij}^{st}$ and previously used $sf\_link\_share_{ij}$ in Appendix A.2.2 is that $sf\_link\_share_{ij}$ is the generic value irrespective of any flow where as $sf\_link\_share_{ij}^{st}$ is the flow specific value. Equation (A46) means that $sf\_link\_share_{ij}^{st}$ is the maximum amount of the capacity the subflow $[o, h] \in A$ of the flow $[s, t] \in K$ can get on the link

$(o, h) \in A$. Equation (A47) ensures that the subflow $[o, h] \in A$ of the flow $[s, t] \in K$ gets the maximum share of the flow capacity on the link $(i, j) \in A$ if the subflow is bottlenecked on that link. In other words, subflows of the flow $[s, t] \in K$ which are bottlenecked on the link $(i, j) \in A$ get equal allocation from that link:

$$sf\_alloc^{st}_{ohij} \leq sf\_link\_share^{st}_{ij} \qquad \forall(o, h) \in A, (i, j) \in A, [s, t] \in K, \tag{A46}$$

$$sf\_alloc^{st}_{ohij} \geq sf\_link\_share^{st}_{ij} - cap_{ij}(1 - sf\_bneck^{st}_{ohij}) \qquad \forall(o, h) \in A, (i, j) \in A, [s, t] \in K. \tag{A47}$$

Appendix A.2.3. Network Flow Fair Allocation (NFF)

In this paragraph, the constraints to compute the allocation gain for the objective function NFF-objective-III are derived. Let $\varphi^{st}_{single-path} \geq 0$ be the allocation of the single path flow $(s, t) \in K$. Equation (A48) refers to if the flow $(s, t) \in K$ is a multipath path flow, then $\varphi^{st}_{single-path} = 0$. Equations (A49) and (A50) make sure that $\varphi^{st}_{single-path}$ is the allocation of the single path flow $(s, t) \in K$. When $(s, t) \in K$ is a single path flow i.e., $flow\_type^{st} = 0$, $\varphi^{st}_{single-path}$ is the end-to-end allocation of that flow i.e., $\varphi^{st}_{single-path} = \varphi^{st}$

$$\varphi^{st}_{single-path} \leq 1 - flow\_type^{st} \qquad \forall(s, t) \in K, \tag{A48}$$

$$\varphi^{st}_{single-path} \leq \varphi^{st} \qquad \forall(s, t) \in K, \tag{A49}$$

$$\varphi^{st}_{single-path} \geq \varphi^{st} - \beta \cdot flow\_type^{st} \qquad \forall(s, t) \in K. \tag{A50}$$

**Network Congestion Groups**

Formulating the equation sets to identify the flows sharing the same congested link requires that all links which are congested for the flows $[s, t] \in K$ are identified. A link is said to be congested only when the link is fully utilized. Consider a variable $unused\_cap_{ij}$ that calculates the unused capacity of the link $(i, j) \in A$ after allocation. If the link $(i, j) \in A$ is congested, then $unused\_cap_{ij} = 0$ for that link:

$$unused\_cap_{ij} = cap_{ij} - \sum_{(q,r) \in K} f^{qr}_{ij} \qquad \forall(i, j) \in A. \tag{A51}$$

The binary variable $y^{st}_{ij}$ identifies at least one congested link for the flow $[s, t] \in K$:

$$\beta \cdot unused\_cap_{ij} + y^{st}_{ij} \geq x^{st}_{ij} \qquad \forall(i, j) \in K, [s, t] \in K. \tag{A52}$$

Let $cong\_flow\_count^{st}_{ij}$ be a counter for the number of flows sharing the same congested link $(i, j) \in A$ along with flow $[s, t] \in K$. Equation (A53) refers to the fact that if the link $(i, j) \in A$ not being congested for the flow $[s, t] \in K$, then the variable $cong\_flow\_count^{st}_{ij} = 0$. Equations (A54) and (A55) make sure that when the link $(i, j) \in A$ is fully utilized for the flow $[s, t] \in K$, $cong\_flow\_count^{st}_{ij}$ counts the number of flows that are congested on that link. When $y^{st}_{ij} = 1$, then $cong\_flow\_count^{st}_{ij} = \sum_{(q,r) \in K} y^{qr}_{ij}$.

$$cong\_flow\_count^{st}_{ij} \leq \beta \cdot y^{st}_{ij} \qquad \forall(i, j) \in K, [s, t] \in K, \tag{A53}$$

$$cong\_flow\_count^{st}_{ij} \leq \sum_{(q,r) \in K} y^{qr}_{ij} \qquad \forall(i, j) \in K, [s, t] \in K, \tag{A54}$$

$$cong\_flow\_count^{st}_{ij} \geq \sum_{(q,r) \in K} y^{qr}_{ij} - cap_{ij} \cdot (1 - y^{st}_{ij}). \tag{A55}$$

When the link $(i, j) \in A$ is not congested for the flow $[s, t] \in K$, then the value of the variable $cong\_group\_id_{ij}^{st}$ is set to zero by Equation (A56). Equation (A57) ensures the variable $cong\_group\_id_{ij}^{st}$ can only be set to 1 only if the congested link $(i, j) \in A$ is shared by other flows along with the flow $[s, t] \in K$. Equation (A58) means that when $cong\_flow\_count_{ij}^{st} \geq 2$, then $cong\_group\_id_{ij}^{st}$ is bound to be 1. This implies that, if the flow $[s, t] \in K$ shares the congested link $(i, j) \in A$ with any other flow, $cong\_group\_id_{ij}^{st}$ is set to 1:

$$cong\_group\_id_{ij}^{st} \leq y_{ij}^{st} \qquad \forall (i, j) \in K, [s, t] \in K, \tag{A56}$$

$$\beta \cdot cong\_group\_id_{ij}^{st} \leq \sum_{(q,r) \in K} y_{ij}^{qr} + \beta - 2 \qquad \forall (i, j) \in K, [s, t] \in K, \tag{A57}$$

$$\beta \cdot cong\_group\_id_{ij}^{st} + 1 \geq cong\_flow\_count_{ij}^{st} \qquad \forall (i, j) \in K, [s, t] \in K. \tag{A58}$$

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
