# Peer review of "A Mathematical Model for Efficient and Fair Resource Assignment in Multipath Transport"

_futureinternet, doi:10.3390/fi11020039_

Reviewer 1 Report

The authors present a mixed (non-)linear programming (MINLP) solution that provides an optimum solution to allocate link capacities in a network to a number of given traffic demands considering both the maximization of link utilization as well as fairness between transport layer data flows or subflows. 

The submission comes timely and is well motivated. Solutions provided are technically solid. Presentation is ok, but more emphasis should be aimed at readership in order to make the paper even more readable. References are mostly relevant and up-to-date capturing the state-of-the-art.

Author Response

We thank the reviewer for the effort and the valuable comments.

Reviewer's comment: more emphasis should be aimed at readership in order to make the paper
even more readable.

Answer: See the updates of the paper marked in yellow.

Reviewer 2 Report

- The proposal is almost appealing and interesting, and the method deserves some consideration.

- The text, in general, reads well, but a grammatical revision could improve it further.

- To improve the paper, I suggest to introduce the following recommendations:

### it is useful that the authors present a real frank account of the strengths and weaknesses of their proposed research method.

### the authors could highlight better the new scientific contribution. What is the main advantage of the paper?

### the conclusion of the article has to be improved. The authors can better discuss their theoretical contributions compared to those in related and cited papers. The authors can also explain the research limitations of their writing and can supply solid, insightful and practical future research suggestions.

### in the introduction, the authors could include a good literature survey to show precisely what is novel about their approach. It would be useful to introduce a clear discussion on the current literature versus the unique contribution of the paper. 

Author Response

We thank the reviewer for the effort and the valuable comments.

Reviewer's comment: The text, in general, reads well, but a grammatical revision could improve it further.
***Answer: Some grammar mistakes, in particular in section 5, were fixed (highlighted in yellow).

Reviewer's comment: ###in the introduction, the authors could include a good literature survey to show precisely what is novel about their approach. It would be useful to introduce a clear discussion on the current literature versus the unique contribution of the paper.
***Answer: see the newly added literature survey on p. 2, lines 47 to 77.

Reviewer's comment: ### the authors could highlight better the new scientific contribution. What is the main advantage of the paper?
***Answer: in the newly provided literature survey mentioned above, it is stated what is the scope of other publications and it is elaborated how
our paper differs from these publications. The scope and novelty of the paper is then again summarized in the already existing text on p. 3 in 
lines 79-83.

Reviewer's comment: ### the conclusion of the article has to be improved. The authors can better discuss their theoretical contributions compared to those in related and cited papers. The authors can also explain the research limitations of their writing and can supply solid, insightful and
practical future research suggestions. 
***Answer: See new text in the conclusion on p. 33, l. 712 to 718. Besides this extension, we believe that the already existing second
paragraph of the conclusion discusses important limitations due to existing fairness methods which are limited to the single-path
transport case amd resulting possible future research aspects.

Reviewer's comment: ### it is useful that the authors present a real frank account of the strengths and weaknesses of their proposed research method.
***Answer: We believe that this issue is covered by the updated literature survey where the scope of the paper is compared to other
publications and be the updated conclusion.

Reviewer 3 Report

No comments

Author Response

We thank the reviewer for the effort.

Round  2

Reviewer 2 Report

The authors have addressed the comments. In my opinion, the paper can be accepted.